# A postzygotic *GNA13* variant upregulates the RHOA/ROCK pathway and alters melanocyte function in a mosaic skin hypopigmentation syndrome

Rana El Masri[1,12], Alberto Iannuzzo [1], Paul Kuentz [2,3], Rachida Tacine[1], Marie Vincent [4], Sébastien Barbarot[5], Fanny Morice-Picard[6], Franck Boralevi[6,15], Naia Oillarburu[7], Juliette Mazereeuw-Hautier [7], Yannis Duffourd[2,8], Laurence Faivre [2,9], Arthur Sorlin [2,10,13], Pierre Vabres[2,10,11,14] ✉ & Jérôme Delon [1,14] ✉

The genetic bases of mosaic pigmentation disorders have increasingly been identified, but these conditions remain poorly characterised, and their pathophysiology is unclear. Here, we report in four unrelated patients that a recurrent postzygotic mutation in *GNA13* is responsible for a recognizable syndrome with hypomelanosis of Ito associated with developmental anomalies. *GNA13* encodes Gα$_{13}$, a subunit of αβγ heterotrimeric G proteins coupled to specific transmembrane receptors known as G-protein coupled receptors. In-depth functional investigations revealed that this R200K mutation provides a gain of function to Gα$_{13}$. Mechanistically, we show that this variant hyperactivates the RHOA/ROCK signalling pathway that consequently increases actin polymerisation and myosin light chains phosphorylation, and promotes melanocytes rounding. Our results also indicate that R200K Gα$_{13}$ hyperactivates the YAP signalling pathway. All these changes appear to affect cell migration and adhesion but not the proliferation. Our results suggest that hypopigmentation can result from a defect in melanosome transfer to keratinocytes due to cell shape alterations. These findings highlight the interaction between heterotrimeric G proteins and the RHOA pathway, and their role in melanocyte function.

Embryonic lethal mutations surviving by mosaicism, which result in deregulation of signalling pathways essential for cell viability, have increasingly been found to be responsible for human development anomalies involving the skin. Hypopigmentation in a typical linear or whorled pattern, also known as hypomelanosis of Ito, or "linear hypomelanosis in narrow bands", has long been recognized as a hallmark of cutaneous mosaicism. Since hyperpigmentation may also be present, making it difficult to determine whether the affected skin is hypopigmented or hyperpigmented, the term "pigmentary mosaicism," encompassing both types of linear dyschromia, is sometimes preferred. Hypomelanosis of Ito has also been reported as "pigmentary mosaicism of the Ito type"[1]. We previously identified postzygotic *MTOR* or *RHOA* mutations in the skin of patients with hypomelanosis of Ito associated with specific extracutaneous anomalies, defining newly recognized mosaic syndromes with pigmentation anomalies[2,3], and supporting a dyadic delineation[4] of pigmentary mosaicism including

both phenotypes and genotypes. These conditions also highlight the major roles of the PI3K-AKT-mTOR and RHOA-ROCK pathways in human embryonic development and melanogenesis. Whereas upregulation of the mTORC1 complex, caused by *MTOR* postzygotic activating mutations, could directly result in cutaneous hypopigmentation through suppression of melanogenesis, little is known about the role of the RHOA/ROCK signalling pathway in the control of melanogenesis. Expression of a constitutively active RHOA mutant inhibits the formation of dendrites in melanocytes[5]. However, the consequences on melanogenesis of the *RHOA* mutations found in patients with pigmentary mosaicism and development anomalies remain unknown.

Here, we have discovered a common mutation in the *GNA13* gene in four unrelated patients with hypomelanosis of Ito and developmental anomalies, clinically reminiscent of patients with RHOA-related mosaic neuroectodermal syndrome. *GNA13* encodes $G\alpha_{13}$, a subunit of a heterotrimeric G protein ($\alpha\beta\gamma$) coupled to specific transmembrane receptors known as G-protein coupled receptors (GPCR)[6,7]. Upon GPCR stimulation, the $G\alpha_{13}$-GTP active form can interact with downstream effectors and kinases to regulate a variety of essential cellular functions such as cytoskeletal modifications, gene transcription, cell migration, and cell division. The signalling process ends with the hydrolysis of GTP, which is mediated by the intrinsic GTPase activity of the Gα subunit.

Here, we have studied the functional consequences of this mutation and propose molecular and cellular mechanisms through which the clinical characteristics appear in patients.

## Results

### Four patients with hypomelanosis of Ito carry the same *GNA13* single nucleotide variation

We evaluated four unrelated individuals with similar phenotypes as part of the MUSTARD cohort. Two patients were initially suspected to be affected with *RHOA* mosaic neuroectodermal syndrome by referring clinicians. Common clinical features included skin linear hypopigmentation in all patients, with either spindle-shaped, guttate, segmental flag-like (triangular or quandrangular) or phylloid hypopigmentation, variably associated with facial asymmetry or asymmetric limb hypoplasia, all of which strongly suggested mosaicism (Fig. 1a–e, Table 1). Although the linear pattern of hypopigmentation was suggestive of Blaschko's lines, none of the patients had typical V-shaped lines of the spinal area, italic S-shaped lines along their limbs, or whorled pattern elsewhere. Three patients had acral anomalies, particularly polysyndactyly in two (Fig. 1f). Hypotrichosis involving hypopigmented skin, hair scalp or eyelashes (partial madarosis) was present in three patients (Fig. 1g). Three patients also had atresia of the gastrointestinal tract, involving either the jejunum or the colon (Table 1). Two patients had delayed cutaneous wound healing for several months on affected areas only (Fig. 1a), and three had ocular anomalies, particularly coloboma of iris or eyelid, or unilateral microphthalmia. One patient had peripheral neuropathy and white matter hyperintensities on MRI, and one had hydrocephalus. None had intellectual impairment or seizures.

Initially, Patient 2 had paired (affected skin and blood) exome sequencing (coverage of 210X in affected skin and 80X in blood). We identified a *GNA13* postzygotic variant (NC_000017.11:g.65014792C>T, NM_006572.6:c.599G>A, NP_006563.2:p.(Arg200Lys)) in affected skin (absent in blood) in 21% of reads (35/163). We then used *GNA13* targeted amplicon sequencing in affected skin in all patients and identified the same *GNA13* postzygotic variant that was supported by 25–36% of reads in hypopigmented skin (Table 1, Supplementary Fig. 1). This *GNA13* variant could only be detected in buccal swabs from two patients (4 and 11 % of reads) and was absent from blood and urines for all tested patients (Table 1).

### The mutated Arg200 is highly conserved

The arginine at position 200 is localised in the switch I region (Fig. 2a, b), which is a flexible region that can bind to GTP or GDP and change its conformation depending on the protein's activation state, allowing $G\alpha_{13}$ to interact with various downstream signalling effectors.

In addition, the presence of an arginine at this position is completely conserved across species from *Drosophila* to humans (Fig. 2c) and among Gα subunits (Fig. 2d). It has been reported that this arginine is critical for GTPase activity in several Gα subunits[8].

### The $G\alpha_{13}$ R200K variant alters cellular morphology and increases actin polymerization

To investigate the cellular impact of the p.(Arg200Lys) mutation identified, we expressed this variant in the B16-F0 melanoma cell line. The effects of the p.(Arg200Lys) mutation were systematically compared to those of a well-characterised artificial constitutively active $G\alpha_{13}$ mutant containing the p.(Glu226Leu) substitution.

We first determined if the mutation affected the cell's cytoskeletal organisation. For this purpose, the network of filamentous actin (F-actin) was visualised and quantified in cells expressing YFP-tagged wild-type (WT) or mutant $G\alpha_{13}$. Cells expressing both mutants have higher F-actin content than WT expressing cells (Fig. 3a, b). Furthermore, actin labelling appears to be more localised in the cell periphery in most mutant cells. The shape of the cells was another intriguing observation. Cells expressing mutants exhibit a smaller perimeter, increased circularity and solidity (indicative of a more regular shape) when compared to WT cells (Fig. 3c–e). Importantly, the same changes in cell morphology and F-actin content induced by $G\alpha_{13}$ R200K and Q226L variants were also observed in the melanoma SK-MEL-28 cell line (Supplementary Fig. 2a–e) and in primary Normal Human Epithelial Melanocytes (NHEM) (Supplementary Fig. 3a).

To ensure that the changes in shape induced by mutant $G\alpha_{13}$ were not caused by dying cells, we performed an annexin V labelling three days after the transfection. The findings confirmed that the two mutations have no effect on the percentage of dying cells (Supplementary Fig. 4).

As a result, we can conclude that the p.(Arg200Lys) mutation causes an increase in actin polymerization and a rounder cell shape, independently of any cell surface receptor engagement. These findings are consistent with the variant being constitutively active, indicating that the mutation found in the patients is a gain-of-function mutation.

### The $G\alpha_{13}$ R200K variant has a broad effect on the cytoskeleton

Beyond actin filaments, we next investigated the status of two other major cytoskeletal proteins: myosin and vinculin.

Non-muscle myosins are proteins that interact with actin to control cell morphology. To determine whether the mutation affects myosin activity, we performed immunocytochemistry labelling of the phosphorylation of the myosin light chains (pMLC), which represent their active form. The results show that both mutants induce an increase in the pMLC cell content in both B16-F0 melanoma (Fig. 4a, b) and primary NHEM (Supplementary Fig. 3b) cells. This suggests that myosin/actin interaction is increased by the pathogenic $G\alpha_{13}$ R200K variant. This is expected to favour cell contractility, which is likely to be responsible for the induction of the round shape of the cells.

Next, we focused our analysis on vinculin which is a component of focal adhesions that links the cytoskeleton to extracellular matrix proteins. Vinculin staining and quantification of the vinculin dots revealed that cells expressing the $G\alpha_{13}$ R200K or Q226L mutants have fewer focal adhesions which are more likely to be localised at the cell periphery than cells expressing the WT form (Fig. 4c, d).

These findings indicate that the p.(Arg200Lys) mutation affects cell cytoskeletal organisation and may disrupt cell adhesion.

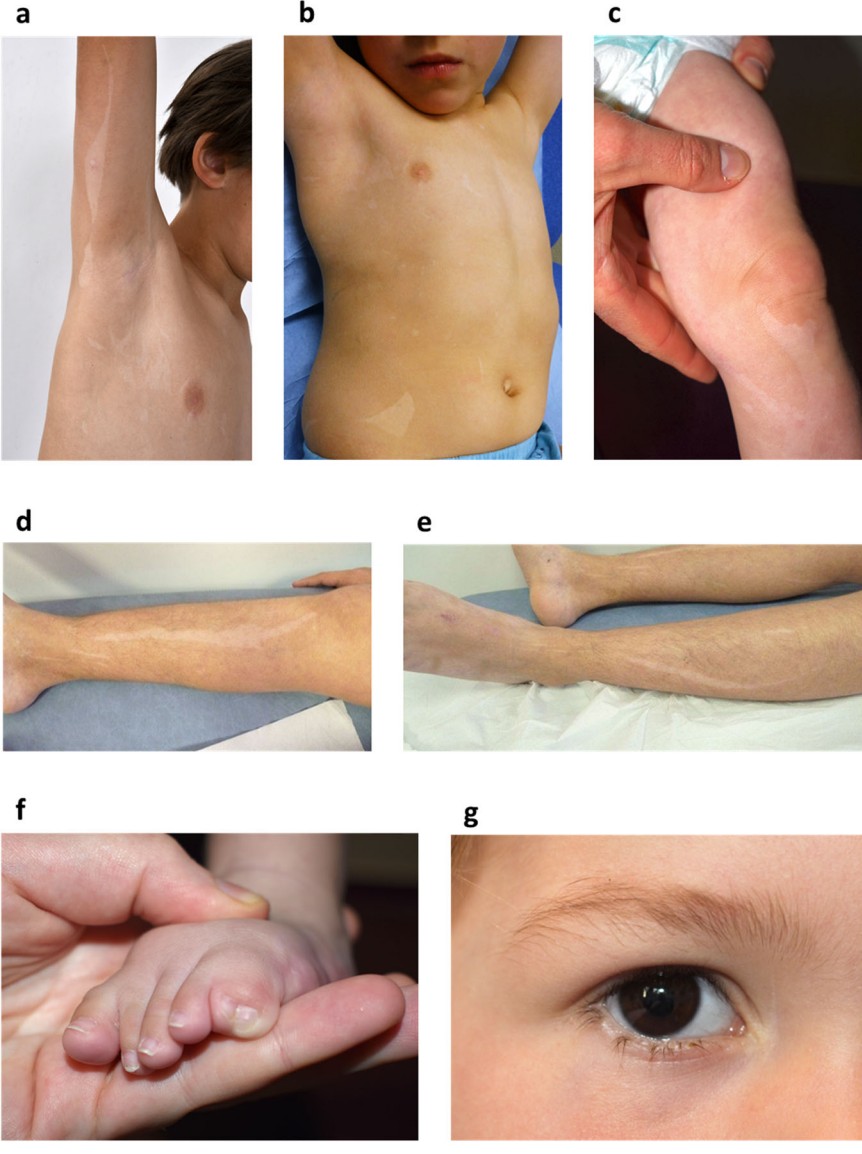

**Fig. 1 | Clinical findings in *GNA13* patients. a–e** Patterns of hypopigmentation in *GNA13* mosaic patients. **a** Phylloid, splash-like (Patient 1, aged 14, also showing biopsy scar on right arm); **b** Flag-shaped (triangular), splash-like (Patient 1, aged 5); **c** Splash-like (Patient 3); **d**, **e** (Patient 2) linear, spindle shaped; **f** Toe polysyndactyly (Patient 3); **g** Patchy hypotrichosis of lower eyelashes (Patient 4).

## Gα₁₃ R200K hyperactivates the RHOA/ROCK signalling pathway

$G\alpha_{13}$ is known to activate RHOA[9], a prominent member of RHO GTPases which are involved in complex diseases in humans[10]. Therefore, we tested the activation state of RHOA in cells transfected with the $G\alpha_{13}$ mutants by quantifying the levels of active GTP-bound RHOA. The results show that at the basal level, independently from any cell surface receptor engagement, R200K and Q226L $G\alpha_{13}$ mutants hyperactivate RHOA in both the B16-F0 melanoma cell line (Fig. 5a) and primary melanocytes (Supplementary Fig. 3c). In parallel, we show that cells expressing a constitutively active form of RHOA (RHOA Q63L) exhibit features similar to those expressing the R200K mutation, regarding the increase in actin polymerization (Fig. 5b) and the round morphology (Fig. 5c).

ROCK1 and ROCK2 are RHOA effectors that have been shown to phosphorylate MLC. To determine whether the cytoskeletal and morphological consequences elicited by the $G\alpha_{13}$ R200K mutation occur via a hyperactivation of the RHOA/ROCK pathway, we pharmacologically inhibited either RHOA with CT04 (Fig. 5d, e) or ROCK with Y27632 (Fig. 5f, g) in cells expressing WT or mutant $G\alpha_{13}$. The inhibition

of RHOA or ROCK had no effect on the F-actin content of cells expressing WT $G\alpha_{13}$, but it severely blocked the increased F-actin polymerization caused by the $G\alpha_{13}$ mutants (Fig. 5d, f). Furthermore, it also reduced the enhanced circularity caused by the mutations and partially restored the cells' original shapes (Fig. 5e, g). Blocking ROCK also resulted in the loss of the mutant-induced enhanced pMLC levels (Supplementary Fig. 5a), confirming that MLC was phosphorylated by ROCK. Interestingly, ROCK inhibition also reversed the morphological effects induced by the mutants as measured by the perimeter and solidity parameters (Supplementary Fig. 5b, c).

Thus, these findings indicate that the functional defects reported above occur via a hyperactivation of the RHOA/ROCK signalling pathway.

## YAP/TAZ signalling pathway is hyperactivated without any effect on cell proliferation

It has previously been shown, in HEK293T cells, that the $G\alpha_{13}$ R200K mutation disrupts the signalling of YAP and its ortholog TAZ[11]. Thus, we have studied this particular pathway in cells of the melanocyte lineage.

**Table 1 | Clinical phenotypes of the four patients carrying the c.599G>A substitution in *GNA13***

| Patient | | Patient 1 | Patient 2 | Patient 3 | Patient 4 |
|---|---|---|---|---|---|
| GNA13 Variant | Nomenclature | chr17:g.63010910C>T NM_006572.4:c.599G>A p.(Arg200Lys) | chr17:g.63010910C>T NM_006572.4:c.599G>A p.(Arg200Lys) | chr17:g.63010910C>T NM_006572.4:c.599G>A p.(Arg200Lys) | chr17:g.63010910C>T NM_006572.4:c.599G>A p.(Arg200Lys) |
| Tissue VAF[a] | Affected skin | 29.0% | 30.0% | 25.0% | 36% |
| | Blood | 0.3% (NS) | 0.00% (NS) | – | 0.0% (NS) |
| | Buccal swab | 0.1% (NS) | 10.6% | 3.9% | 0.5% (NS) |
| | Urine | 0.0% (NS) | – | 0.1% (NS) | 0.0% (NS) |
| Sex | | Male | Male | Female | Female |
| Age at last visit | | 14 y. 7 mo. | 24 y. | 22 mo. | 9 y. 7 mo. |
| Growth | OFC (cm) | 50.5 (-1 SD) | – | – | – |
| | Height (cm) | 108.5 (-0.5 SD) | – | – | 123 (+0.5 SD) |
| | Weight (kg) | 19.0 (M) | – | – | 22 (M) |
| Asymmetry | | Right-sided hemihypotrophy (lower limb) | Right-sided hemifacial hypoplasia | No | Right-sided hemifacial hypoplasia |
| Craniofacial anomalies | | | Lower limb asymmetry Plagiocephaly Scoliosis | | |
| Pregnancy and birth | Fetal ultrasonography | – | Intrauterine growth retardation | – | Normal |
| | Term | 41 wk | – | – | 39 wk |
| | OFC at birth (cm) | NR | – | – | 33.5 |
| | Length at birth (cm) | 44.0 | – | – | 49.5 |
| | Weight at birth (g) | 2500 | – | – | 3560 |
| Skin and hair | Hypopigmentation pattern | Linear, spindle-shaped, flag-like, splash-like, phylloid | Linear, spindle-shaped | Linear, splash-like | Linear, spindle-shaped |
| | Location of hypopigmentation | right upper and lower limbs, trunk | Lower limbs bilaterally, face (chin) | Left lower limb | Right upper limb |
| | Hair anomalies | Absence of hair on hypopigmented skin | Patchy alopecia of scalp hair and right eyelashes | NR | Patchy alopecia on parietal scalp and right eyelashes |
| | Wound healing | Delayed wound closure (9 months) after skin biopsy (right upper limb) | Delayed wound closure after skin biopsy (1 month) and after excision of supernumerary digit of the right hand (2–3 months) | NR | NR |
| Neurology | Peripheral nerves | Right foot dysaesthesia and weakness | NR | NR | NR |
| | Brain MRI | White matter hyperintensities (centrum semiovale, right parietal lobe and temporoparietal junction, left frontal lobe) | Hydrocephalus | Normal (at age 6 mo.) | Not performed |
| Acral anomalies | | Camptodactyly | Right hand polysyndactyly Left hand brachymetacarpy Bilateral pes planus (partial talocalcaneal fusion) Toe syndactyly (right foot) | Left foot polysyndactyly | No |
| Ocular anomalies | | No | Coloboma of upper eyelid, bilateral abnormal fundus | Mild iris coloboma | Amblyopia with mild microphalmia of right eye Narrow palpebral fissure of right eye Normal eye fundus |
| Dental anomalies | | No | Odontogenic cysts, dental anomalies | NR | Conical teeth |
| Hearing loss | | No | Mixed hypoacusis of right ear (hearing aid) Otosclerosis Abnormal inner auditory and cochlear nerve canals Cochlear nerve agenesis or hypoplasia | Left ear hypoacusis Chronic otitis media | Left ear hypoacusis |
| Gastrointestinal tract anomalies | | No | Jejunal atresia | Colonic atresia (ascending colon) | Colonic atresia |
| Urinary tract anomalies | | No | Bilateral hydronephrosis, right megaureter | No | No |

VAF variant allele frequency, NR not reported, – missing data, NS not significant (below background noise threshold).
[a]Targeted ultradeep sequencing (TUDS) on affected skin.

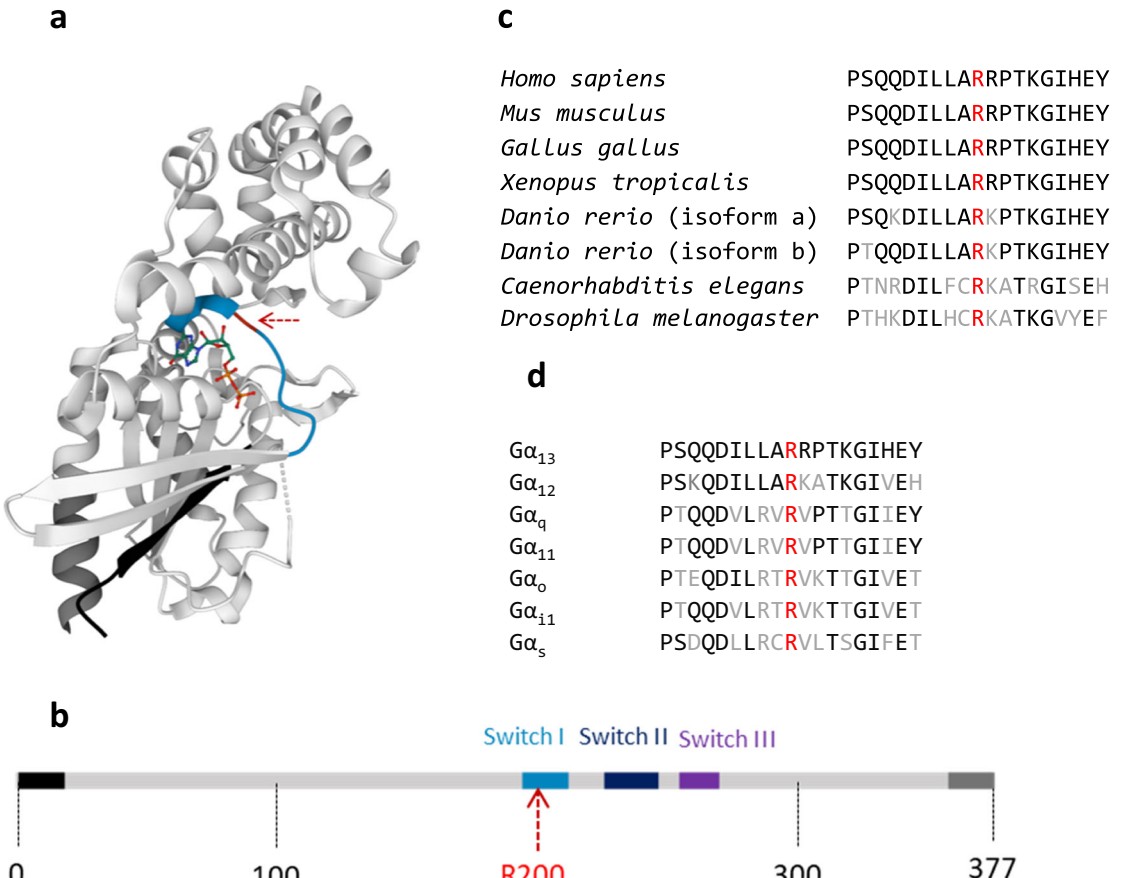

**c**

| | |
|---|---|
| *Homo sapiens* | PSQQDILLA**R**RPTKGIHEY |
| *Mus musculus* | PSQQDILLA**R**RPTKGIHEY |
| *Gallus gallus* | PSQQDILLA**R**RPTKGIHEY |
| *Xenopus tropicalis* | PSQQDILLA**R**RPTKGIHEY |
| *Danio rerio* (isoform a) | PSQKDILLA**R**KPTKGIHEY |
| *Danio rerio* (isoform b) | PTQQDILLA**R**KPTKGIHEY |
| *Caenorhabditis elegans* | PTNRDILF C**R**KATRGISEH |
| *Drosophila melanogaster* | PTHKDILH C**R**KATKGVYEF |

**d**

| | |
|---|---|
| Gα₁₃ | PSQQDILLA**R**RPTKGIHEY |
| Gα₁₂ | PSKQDILLA**R**KATKGIVEH |
| Gα_q | PTQQDVLRV**R**VPTTGIIEY |
| Gα₁₁ | PTQQDVLRV**R**VPTTGIIEY |
| Gα_o | PTEQDILRT**R**VKTTGIVET |
| Gα_i1 | PTQQDVLRT**R**VKTTGIVET |
| Gα_s | PSDQDLLRC**R**VLTSGIFET |

**Fig. 2 | Position of the Gα₁₃ R200K mutation identified in patients. a** 3D X-ray structure of Gα₁₃[35]. The N-terminal, C-terminal and switch I domain are represented in black, dark grey and blue respectively, while the R200 residue is indicated with a red dotted arrow. **b** Linear representation of Gα₁₃ showing the position of the R200 residue. The same colour code used in part **a** is also shown here. **c** Alignment of Gα₁₃ Switch I and surrounding region in different species showing in red the fully conserved arginine which corresponds to Gα₁₃ R200. Identical residues are represented in black and the different ones in grey. **d** Alignment of Gα Switch I and surrounding region showing in red the fully conserved arginine that corresponds to Gα₁₃ R200. Identical residues are represented in black and the different ones in grey.

We first quantified the basal expression levels of YAP and TAZ in the melanocyte cells used in our study (Fig. 6a). We show that B16-F0, SK-MEL-28 and primary NHEM cells all express YAP and TAZ, although at various levels. YAP expression is highest in B16-F0, followed by NHEM, and finally by SK-MEL-28, which exhibit very low YAP levels. These differences in YAP/TAZ co-expression are fully consistent with a previously published study[12]. Expression of WT or mutant Gα₁₃ in these three cell types did not affect their YAP and TAZ protein levels (Supplementary Fig. 8).

Next, we studied the effect of the Gα₁₃ R200K mutation on YAP/TAZ activity by measuring YAP/TAZ nuclear translocation. All three cell types showed strong YAP nuclear localization upon expression of either R200K or Q226L Gα₁₃ mutants (Fig. 6b, Supplementary Figs. 6a and 7a). By contrast, they failed to induce TAZ nuclear translocation (Fig. 6c and Supplementary Fig. 7b). Furthermore, in order to test whether the YAP signalling pathway was functional upon expression of gain-of-functions Gα₁₃ mutants, we next tested whether *ARPC5*, a specific target gene of YAP[12] involved in actin polymerization, was induced in our conditions. Our results show that ARPC5 transcripts were upregulated by both Gα₁₃ R200K and Q226L (Supplementary Fig. 7c), indicating that the YAP signalling pathway is hyperactivated upon expression of gain-of-functions Gα₁₃ mutants in melanocytes.

We also assessed the effects of the Gα₁₃ R200K and Q226L mutations on cell proliferation using CellTrace Violet dilution and flow cytometry analysis. Our results revealed that both mutations have no effect on proliferation in the three cell types (Fig. 6d, Supplementary Figs. 6b and 7d).

Altogether, these results indicate that the R200K Gα₁₃ mutant, identified here in these patients affected by this hypopigmentation syndrome, triggers hyperactivation of the RHOA-dependent transcription mediated by the Hippo pathway effector YAP in melanocytes.

## Gα₁₃ R200K inhibits cell migration

Next, we aimed to determine if the mutations had any effect on cell migration.

We chose the SK-MEL-28 melanoma cell line because, unlike B16-F0 cells, we managed to transduce it with Gα₁₃ with about 80 % efficiency. We then used a scratch wound assay to track the ability of Gα₁₃-expressing cells to migrate and close the wound within two days after the scratch. The results show that non-infected control cells and cells expressing WT Gα₁₃ have regular and comparable migratory abilities (Supplementary Fig. 2f). However, the migration of cells expressing either R200K or Q226L mutants was very much hampered. Altogether, these results indicate that the morphological alterations elicited by the pathogenic R200K Gα₁₃ variant are responsible for a defect in cell migration.

## Gα₁₃ R200K inhibits melanosome transfer to keratinocytes

Finally, to determine the mechanism of skin pigmentary changes exhibited by patients carrying the R200K Gα₁₃ variant, we also studied the impact of this mutation on melanin expression and transfer.

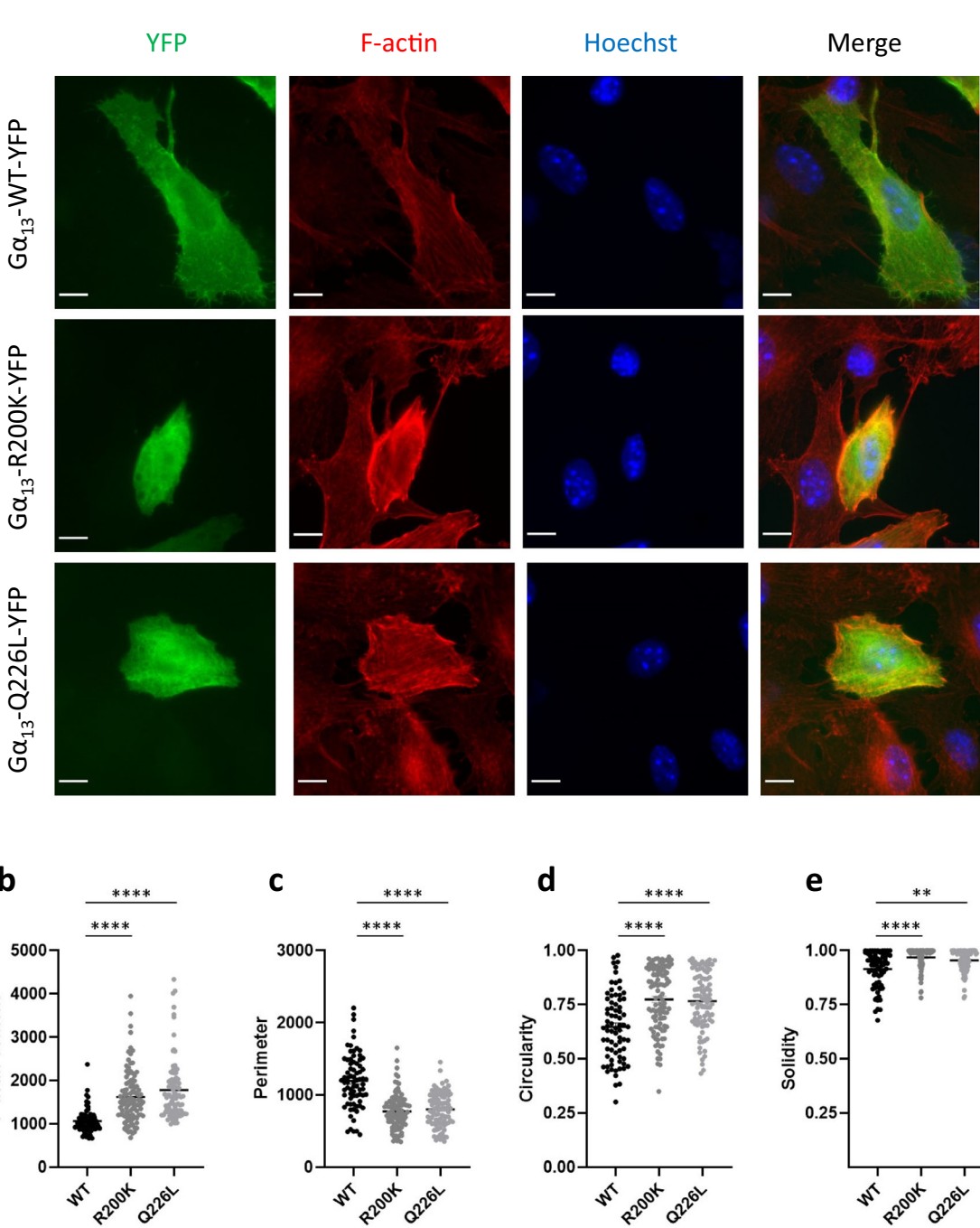

**Fig. 3 | Effect of the Gα₁₃ R200K mutation identified in patients on actin organisation and cell morphology. a** Immunofluorescence images of Gα₁₃ WT-YFP, Gα₁₃ R200K-YFP or Gα₁₃ Q226L-YFP (green) expressed in B16-F0 cells. The cells were labelled with phalloidin for visualising F-actin (red), and Hoechst for nuclei (blue). The scale bar is 10 μm. **b** Quantification of F-actin content that corresponds to the intensity of the red signal. Quantification of cell morphology parameters including perimeter (**c**), circularity (**d**) and solidity (**e**). Means +/− SEM are shown from 75, 103 and 92 Gα₁₃ WT, R200K and Q226L expressing B16-F0 cells, respectively. ANOVA tests were performed. **P = 0.0014; ****P < 0.0001. All graphs shown are representative of eight independent experiments.

The skin pigmentation process occurs through two important steps: (i) production of melanin by the melanocytes in vesicles called melanosomes which undergo maturation steps, and (ii) transfer of melanosomes from melanocytes to keratinocytes. Because the latter step is dependent on the cytoskeleton and the morphology of the melanocytes, we hypothesised that the R200K-induced changes in melanocyte shape could thus disrupt the transfer of melanosomes to keratinocytes.

Melanin Stimulating Hormone (MSH) is a hormone that induces the maturation of melanosomes and their transfer to keratinocytes through some morphological changes in the melanocytes. We show that MSH stimulation of B16-F0 cells expressing WT Gα₁₃ allows them to acquire long and numerous cytoplasmic extensions like dendrites (Fig. 7a). However, the formation of these extensions is very much inhibited in cells expressing mutant Gα₁₃, as shown by the decrease in both numbers (Fig. 7b) and lengths (Fig. 7c) of these acquired

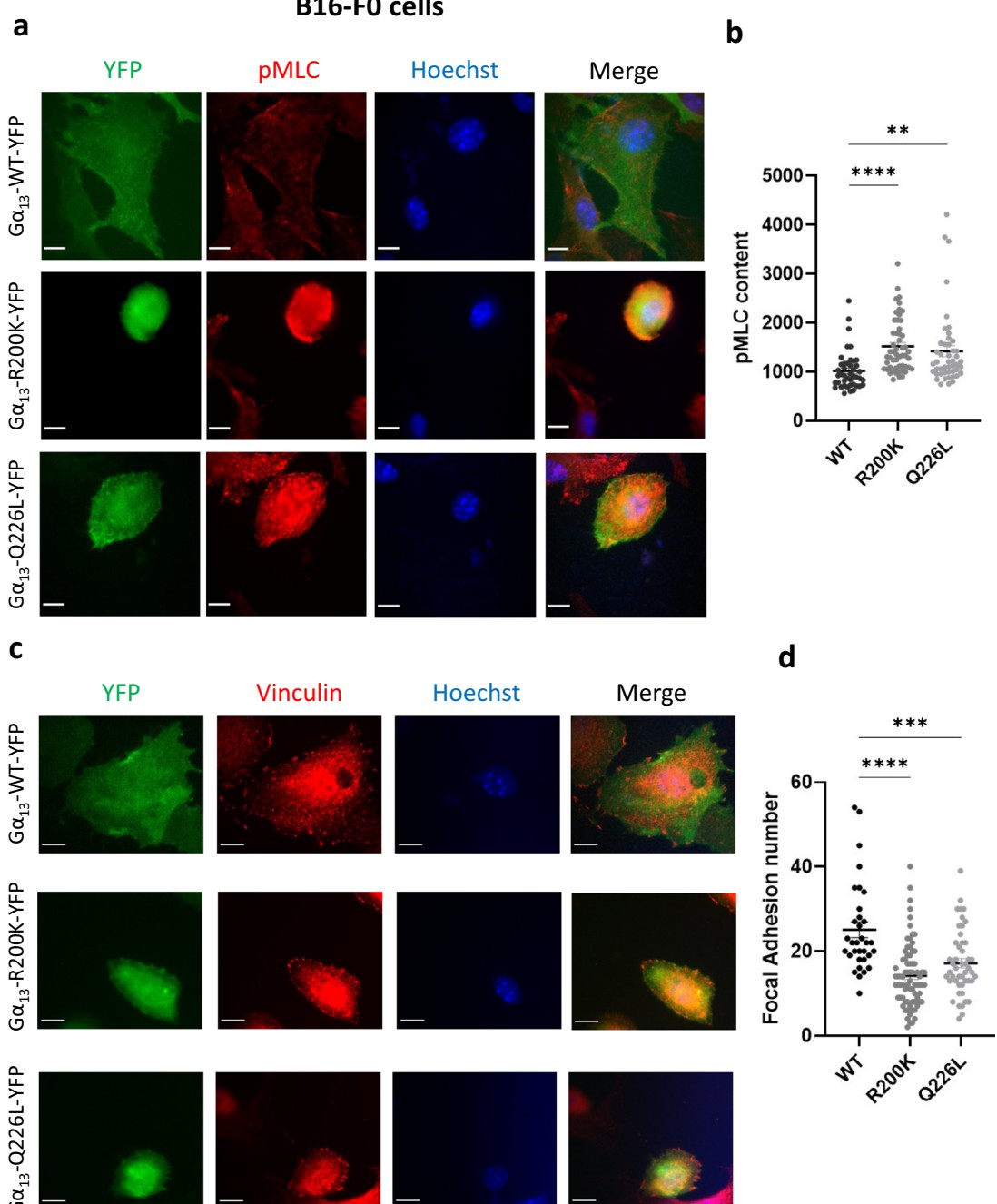

**Fig. 4 | Effect of the Gα13 R200K mutation identified in patients on cytoskeletal proteins.** Immunofluorescence images of Gα13 WT-YFP, Gα13 R200K-YFP or Gα13 Q226L-YFP (green) expressed in B16-F0 cells. The cells were labelled with anti-pMLC (red) (**a**) or anti-vinculin (red) (**c**) antibodies, and Hoechst for nuclei (blue). The scale bar is 10 μm. **b** Quantification of the pMLC content that corresponds to the red signal intensity mean. Means +/− SEM are shown from 43, 53 and 47 Gα13 WT, R200K and Q226L expressing B16-F0 cells, respectively. ANOVA tests were performed. **P = 0.0022; ****P < 0.0001. Graph shown is representative of six independent experiments. **d** Quantification of the number of focal adhesions per cell that corresponds to the mean number of the red dots in each cell. Means +/− SEM are shown from 33, 76 and 48 Gα13 WT, R200K and Q226L expressing B16-F0 cells, respectively. ANOVA tests were performed. ***P = 0.0004; ****P < 0.0001. Graph shown is representative of three independent experiments.

extensions in cells expressing the R200K or Q226L Gα13 variants. Interestingly, these Gα13 mutations do not affect the amount of melanin pigment in melanocytes (Supplementary Fig. 9).

In order to test whether these extensions affect the transfer of melanosomes to keratinocytes, the B16-F0 melanocytes expressing Gα13 WT or mutants were co-cultured with HaCaT keratinocytes and stimulated with MSH (Fig. 7d). The results show that both Gα13 mutations negatively affect the number of transferred melanosomes in keratinocytes, up to the level where these mutants exert a complete

repression of the MSH effect on melanosomes transfer (Fig. 7e). These important results therefore favour an inhibition of melanosomes transfer as the main explanation for the pathogenicity of the R200K Gα13 variant.

## Discussion

Here, we have identified a recurrent postzygotic mutation in *GNA13* encoding Gα13 in patients with cutaneous pigmentary mosaicism and developmental disorder. These findings highlight the importance of

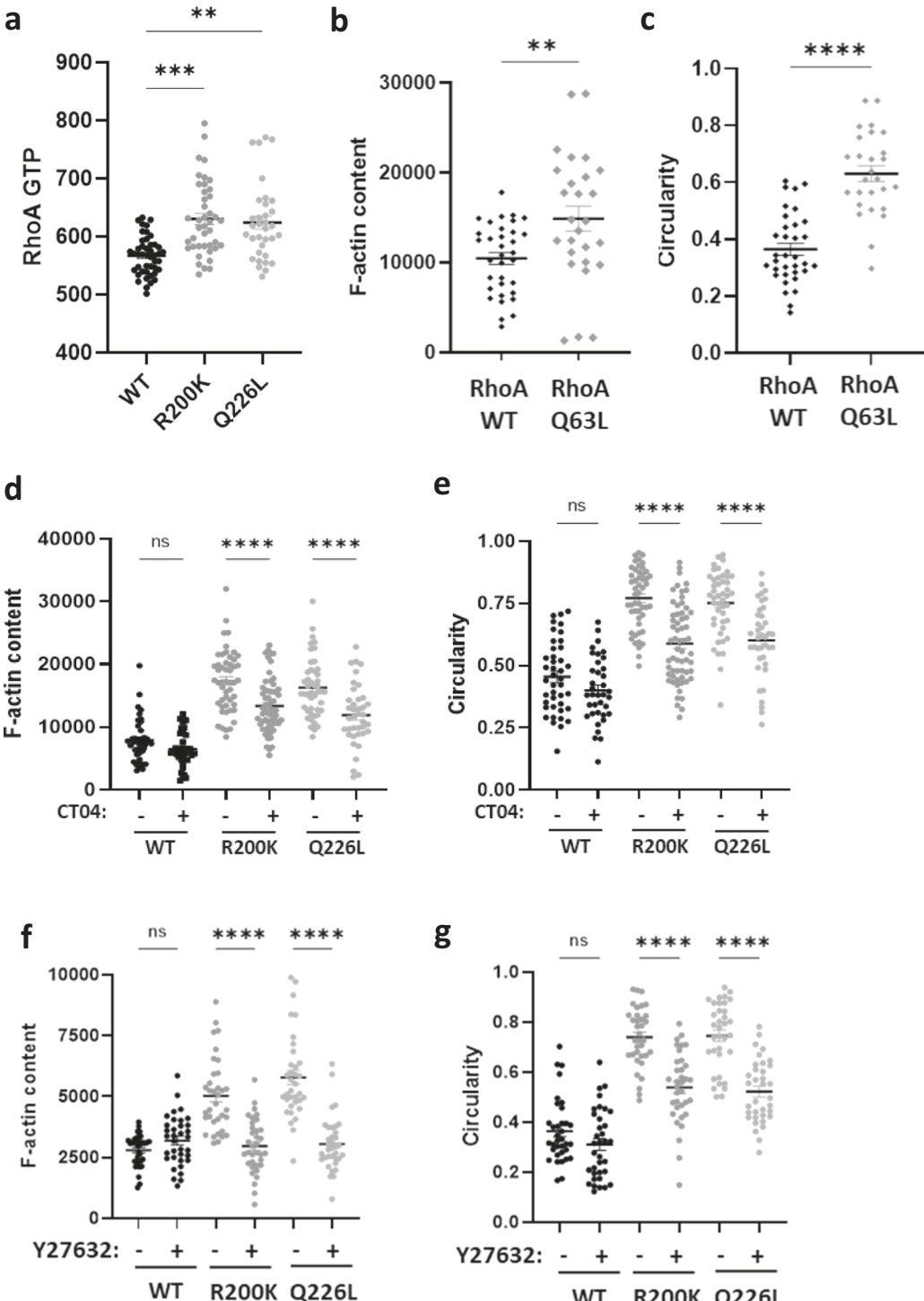

**Fig. 5 | Effect of the Gα₁₃ R200K mutation identified in patients on RHOA signalling. a** Quantification of RHOA-GTP levels in Gα₁₃ WT-YFP, Gα₁₃ R200K-YFP or Gα₁₃ Q226L-YFP expressing B16-F0 cells that correspond to the mean intensity of the signal generated from anti-GTP RHOA antibody labelling. Means +/- SEM are shown from 45, 42 and 34 Gα₁₃ WT, R200K and Q226L expressing B16-F0 cells, respectively. ANOVA tests were performed. **P = 0.0058; ***P = 0.0006. Graph shown is representative of four independent experiments. Quantification of F-actin content (**b**) and circularity (**c**) in RHOA WT-GFP and RHOA Q63L-GFP expressing B16-F0 cells. Means +/− SEM are shown from 34 and 27 RHOA WT and Q63L expressing B16-F0 cells, respectively. ANOVA tests were performed. **P = 0.0058; ****P < 0.0001. All graphs shown are representative of three independent

experiments. Quantification of F-actin content (**d**) and circularity (**e**) in Gα₁₃ WT-YFP, Gα₁₃ R200K-YFP or Gα₁₃ Q226L-YFP expressing B16-F0 cells, upon RHOA inhibition treatment with CT04. Quantification of F-actin content (**f**) and circularity (**g**) in Gα₁₃ WT-YFP, Gα₁₃ R200K-YFP or Gα₁₃ Q226L-YFP expressing B16-F0 cells, upon ROCK inhibition treatment with Y27632. For CT04 treatment, means +/− SEM are shown from 42, 50 and 46 Gα₁₃ WT, R200K and Q226L untreated expressing B16-F0 cells, respectively, and from 37, 61 and 39 Gα₁₃ WT, R200K and Q226L treated expressing B16-F0 cells, respectively. For Y27632 treatment, means +/−SEM are shown from 35 cells of each condition. ANOVA tests were performed. ****P < 0.0001. All graphs shown are representative of three independent experiments.

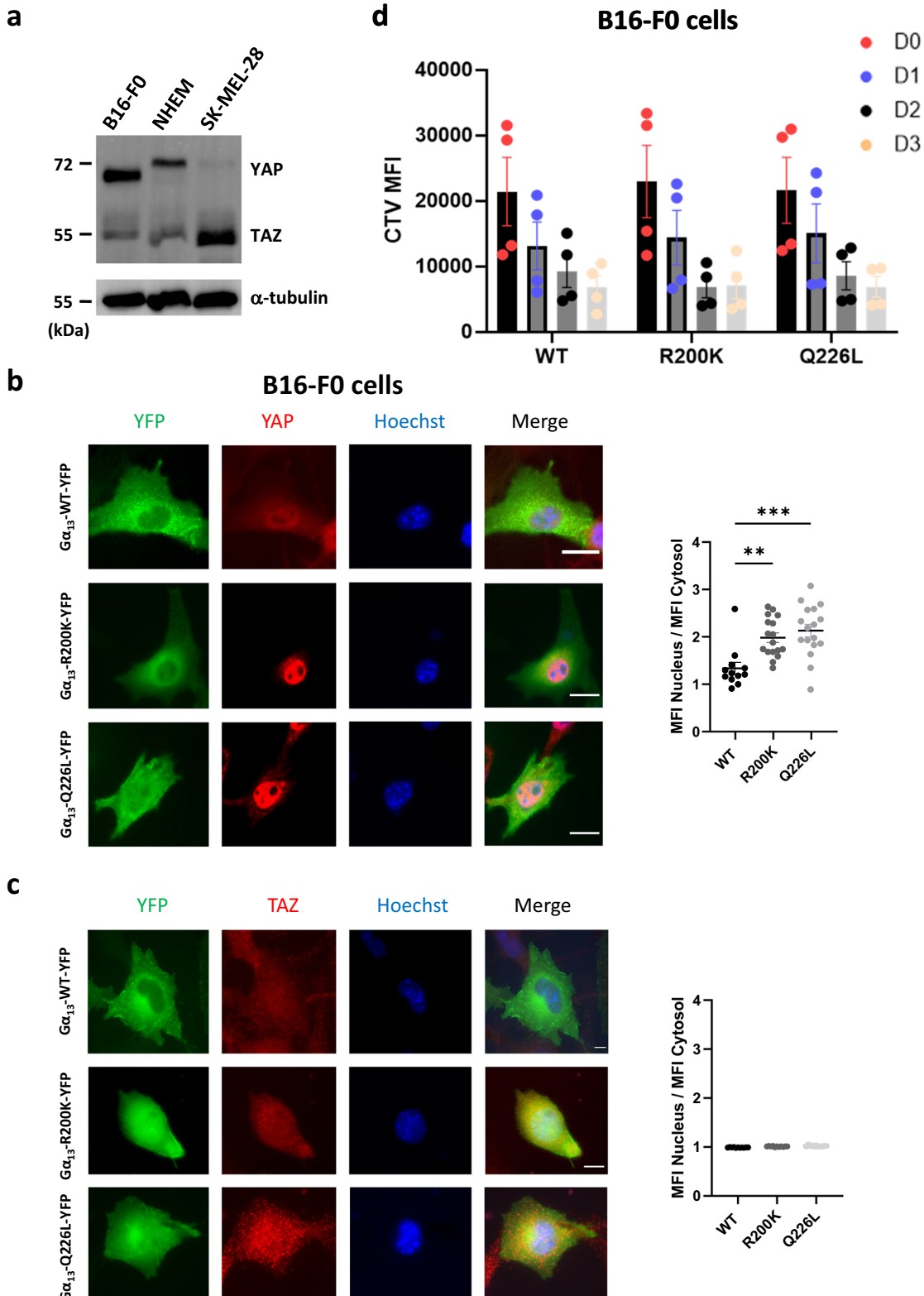

**Fig. 6 | Effect of the Gα₁₃ R200K mutation on YAP/TAZ signalling and proliferation. a** Western blot analysis of YAP and TAZ expression in B16-F0, NHEM and SK-MEL-28 cells. Left: Immunofluorescence images of Gα₁₃ WT-YFP, Gα₁₃ R200K-YFP or Gα₁₃ Q226L-YFP (green), YAP (**b**) or TAZ (**c**) (red) and the nucleus (Hoechst, blue) in B16-F0 cells. Right: Single cell quantifications of the ratios of YAP (**b**) or TAZ (**c**) Mean Fluorescence Intensity (MFI) in the nucleus over the cytosol. Means +/− SEM are shown from 10 to 20 cells of each condition. ANOVA tests were performed. **P = 0.0014; ***P = 0.0001. **d** Flow cytometry quantification of the CellTrace violet mean fluorescence intensity (CTV MFI) at days 1, 2 and 3 following CTV staining of B16-F0 cells at day 0. The dye dilution within days traces the multiple generations of the proliferating cells. Means +/− SEM from four independent experiments are shown.

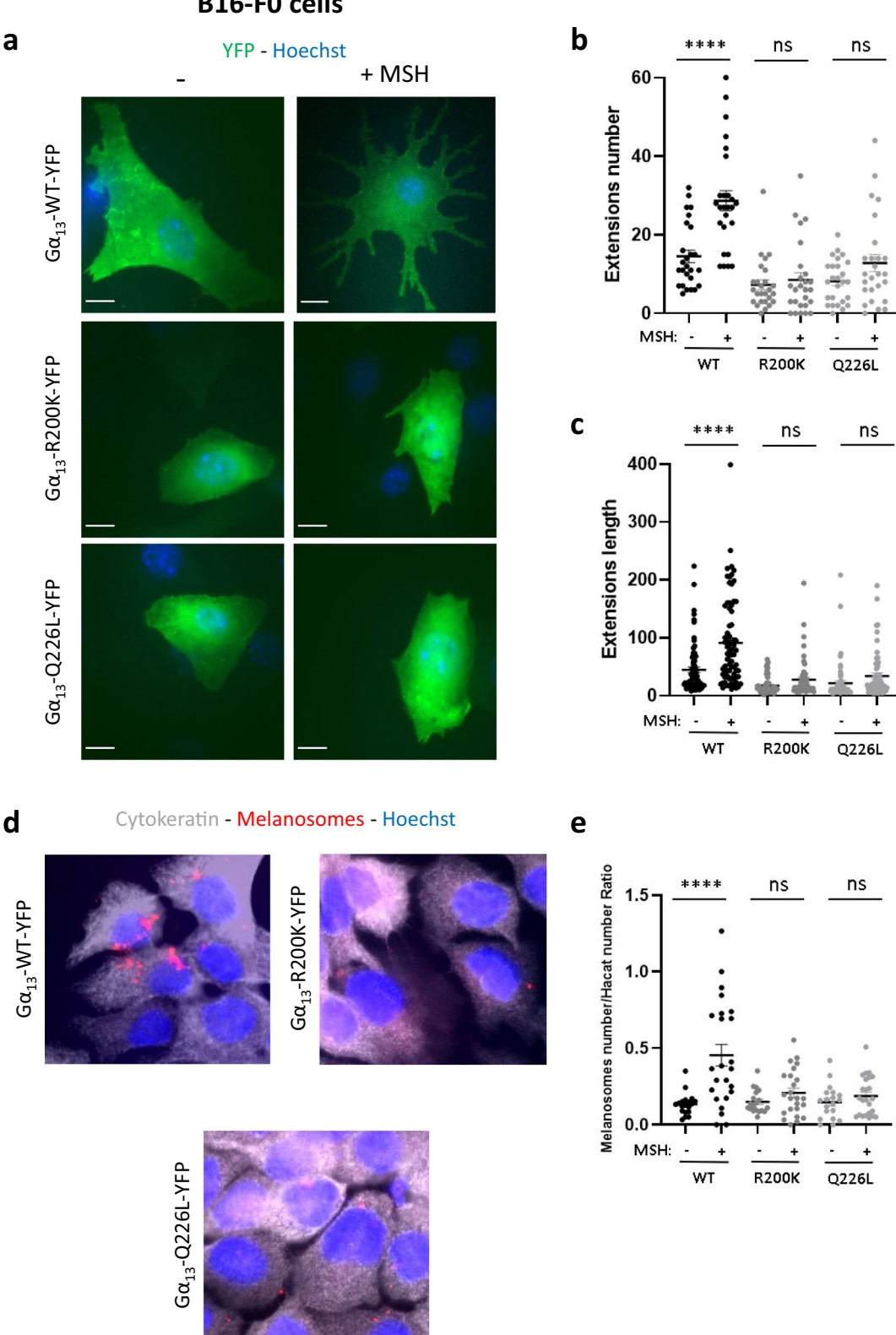

**Fig. 7 | Effect of the Gα₁₃ R200K mutation on melanosomes transfer to keratinocytes. a** Immunofluorescence images of Gα₁₃ WT-YFP, Gα₁₃ R200K-YFP or Gα₁₃ Q226L-YFP (green) expressed in B16-F0 cells in the absence or presence of MSH stimulation. Nuclei are stained with Hoechst (blue). The scale bar is 10 µm. The number of extensions (**b**) of the cells were counted and their length was also measured (**c**). Means +/− SEM are shown from 26 cells of each condition. **d** Immunofluorescence images of HaCaT keratinocytes co-cultured with Gα₁₃ WT-YFP, Gα₁₃ R200K-YFP or Gα₁₃ Q226L-YFP expressing B16-F0 cells upon MSH

stimulation. Keratinocytes are visualised by anti-cytokeratin antibody (white). Melanosomes are stained with anti-TRP1 antibody (red) and nuclei are stained by Hoechst staining (blue). **e** Quantification of melanosomes number (red dots) per HaCaT cells. Means +/− SEM are shown from 19 and 24 cells of each condition, in the absence or presence of MSH, respectively. ANOVA tests were performed. ****P < 0.0001. All graphs shown are representative of three independent experiments.

$G\alpha_{13}$ in human development and provide new insights on its role in melanogenesis. Indeed, linear or patchy hypopigmentation appears as a direct consequence of this *GNA13* mutation on melanin transfer. Although a consistent phenotype cannot be determined from four patients, some clinical features appear relevant among these patients apart from pigmentary mosaicism, particularly delayed wound healing and gastrointestinal (GI) tract atresia which appear to be non-random manifestations in our patients. Defects in cell migration could account for delayed wound healing. Non-syndromic GI tract atresia could also be caused by mosaic *GNA13* mutations restricted to the gut tissue. We believe that many sporadic birth defects are likely to be due to such localised mosaicism. This hypothesis could be tested on tissues obtained from surgical procedures. Consistent with that hypothesis, a search of the Human Protein Atlas indicates that $G\alpha_{13}$ is highly expressed in the GI tract.

To investigate the impact of this mutation, we chose primary melanocytes and melanocytic cell lines to express the mutation. Indeed, direct study of patient skin samples is limited by ethical and medical considerations - such as delayed wound healing observed after skin biopsy in two patients - to justify less invasive explorations. In addition, both the potential loss of the mutation in primary cell cultures derived from the biopsy in affected skin and the observed absence of $G\alpha_{13}$ variant in blood, as already observed in *RHOA* mutated patients[3], confirms the importance of $G\alpha_{13}$/RHOA in hematopoietic stem cell and lymphocyte development, suggesting negative selection of mutant blood cells. However, the mutation was detected in the buccal epithelium of two out of four patients. As a result, buccal smears provide a less invasive method for diagnosing mutations in $G\alpha_{13}$-related mosaic disorders. If that is negative, diagnosis should be performed on a biopsy from affected skin.

From a mechanistic point of view, we demonstrate that the R200K $G\alpha_{13}$ variant found in patients is an activating mutation that alters the cytoskeleton and morphology of cells via hyperactivation of the RHOA/ROCK signalling pathway (Supplementary Fig. 10). This is in line with early literature that showed that expression of a constitutively active RHOA mutant inhibits the formation of dendrites in melanocytes[5]. Here, although our single-cell approach shows an intrinsic degree of variability between individual cells, we report broad shape alterations which inhibit cell adhesion and migration, and the transfer of melanosomes towards keratinocytes but not the amount of melanin produced per se. The mechanisms responsible for melanin transfer from melanocytes to keratinocytes remain quite controversial as four different models have been proposed[13]. However, all these models agree that membrane remodelling and close contacts between melanocytes and keratinocytes are required for melanin transfer. The mechanism of hypopigmentation in hypomelanosis of Ito may vary depending on the gene involved[14]. In patients with MTOR-related pigmentary mosaicism, we found a decrease in intrakeratinocytic melanosomes, and a defect in maturation of melanosomes in melanocytes[2]. Here, the pigmentation defect likely results from defects in intercellular contacts and communication between melanocytes and keratinocytes. However, our results showing an effect of the mutation on cell migration could also suggest that melanocytes are unable to migrate correctly from the neural crest to the epidermis during embryogenesis.

Our data is consistent with a previous report in cancer-associated mutations indicating that the $G\alpha_{13}$ R200K variant is constitutively loaded with GTP, and consequently hyperactive[11]. It was indeed revealed that the R200K mutation is a naturally occurring hotspot mutation in bladder cancer that causes transcriptional upregulation of YAP/TAZ and MRTF-A/B via a RHOGEF - RHOA GTPase cascade. YAP/TAZ are effectors of the Hippo pathway which act as transcriptional coactivators in response to a plethora of environmental clues and can drive metastasis and tumour progression in a variety of cancers[15–17], including in melanoma[18]. Maziarz et al. found that $G\alpha_{13}$ R200K upregulates YAP/TAZ in HEK293T but not in NIH3T3[11]. However, the $G\alpha_{13}$

R200K variant's behaviour in the melanocyte lineage remains unknown. Here, we show that the R200K $G\alpha_{13}$ mutant induces YAP, but not TAZ, nuclear translocation causing its activation and upregulation of the *ARPC5* gene involved in actin polymerization and migration. YAP/TAZ nuclear accumulation has previously been linked to the activation of the RHOA pathway[19]. Thus, the R200K $G\alpha_{13}$ variant activates the YAP signalling pathway most likely through RHOA hyperactivation. We also show a specific effect of YAP over TAZ, in line with other reports showing that both proteins are not necessarily functionally redundant, as recently shown in melanoma[12]. Interestingly, a specific role for YAP has previously been shown in skin regeneration. Specific silencing of YAP in fibroblasts or pharmacological inhibition of YAP/TAZ has been shown to increase dermal regeneration[20]. Thus, an increase of YAP signalling mediated by R200K $G\alpha_{13}$ in melanocytes suggests an inhibition of dermal regeneration that could occur in the patients we describe here, potentially explaining the wound healing defects observed for Patients 1 and 2.

Contrary to the pro-oncogenic role of the activatory $G\alpha_{13}$ R200K mutation in bladder cancer, B cell lymphomas have been reported to be driven by inhibitory mutations in *GNA13* or *RHOA*[21,22]. Here, we show no effect of both gain-of-function R200K and Q226L $G\alpha_{13}$ mutations on melanocytic cell line proliferation. The presence of an oncogenic mutation in our patients could raise the issue of a possible cancer predisposition. However, the risk of neoplasia or haematological malignancy in patients with *GNA13* mosaicism is unknown. To our knowledge, no increased risk of cancer has been reported in hypomelanosis of Ito. Moreover, many other mosaic developmental syndromes are caused by activating postzygotic mutations in oncogenes, yet the overall incidence of cancer remains low, as reported in *PIK3CA*-related overgrowth syndrome[23].

Other genes encoding other isoforms of Gα protein subunits are also involved in both YAP-induced uveal melanoma[24–26] and mosaic disorders in humans when mutated, for example *GNAQ* and *GNA11* in the Sturge-Weber syndrome[27], or *GNAS* in the McCune-Albright syndrome[28,29]. Interestingly, it is the same homologous arginine mutation - conserved among these Gα isoforms - that is found in the postzygotic state in these patients, although no mechanistic explanation has been provided so far. Based on the results we present here, we propose that the signalling pathways downstream of these Gα proteins could, by crosstalk with RHO GTPases, also affect the activation of RHOA.

We previously identified postzygotic *RHOA* mutations as the genetic basis of a mosaic neuroectodermal syndrome with hypomelanosis of Ito. Clinically, their manifestations overlap with those of patients with a postzygotic *GNA13* mutation. Our findings on the functional consequences of this *GNA13* mutation add new evidence and strongly support the pivotal role of the RHOA/ROCK pathway in human developmental disorders with pigmentary mosaicism[3]. Interestingly, these *RHOA* mutations are located in the switch regions of *RHOA* and appear to be inactivating, resulting in a decrease in MLC phosphorylation. Although these observations may appear discordant with the results reported here, they could rather highlight the significance of the cycle between active and inactive forms of RHOA depending on cell needs, and that deregulation of this balance towards either form, even if affecting the downstream signalling pathways in opposite ways, can result in similar phenotypes. Dysregulation of the RHOA/ROCK pathway may also be directly involved in abnormal skin wound healing, as well as in gastrointestinal tract atresia. Indeed, in mouse skin, RhoA has been found to upregulate activin B-induced wound healing, where activin B stimulates proliferation of keratinocytes and hair follicle cells[30]. Likewise, RHOA inactivation-induced inhibition of intestinal epithelial cell migration[31] could explain gastrointestinal tract atresia.

To summarise, the discovery of *GNA13* as a gene associated with pigmentary mosaicism with development anomalies could reveal common pathogenic pathways, improving our understanding of their

causes and, ultimately, suggesting that RHOA or ROCK inhibition, which restored some of the cell morphology and cytoskeletal organisation alterations caused by the mutation, could be used as a therapeutic approach in mosaic developmental disorders with pigmentation anomalies.

## Methods

### Study participants

This study includes four unrelated, affected children and their unaffected parents. Individuals were recruited and phenotyped by dermatologists and geneticists collaborators of the M.U.S.T.A.R.D. (Mosaic Undiagnosed Skin Traits And Related Disorders) cohort whose ethical approval was given by the Ethics Committee of the *Fédération Hospitalo-Universitaire* TRANSLAD (*Médecine TRANSLAtionnelle dans les anomalies du Développement*) from the *Centre Hospitalier Universitaire* Dijon Bourgogne, as part of a French collaborative effort to identify genes involved in skin mosaic syndromes. Inclusion criteria consisted of skin hypopigmentation in a mosaic pattern associated with any type of extracutaneous involvement. Informed written consent was obtained from all subjects and participating family members. The authors affirm that the research participants or their parents/legal guardians provided written informed consent for publication of the images in Fig. 1 and the potentially identifiable clinical data in Table 1.

### Next generation sequencing

Sequencing was performed directly on fresh biopsy from affected (hypopigmented) skin, since cultured fibroblasts samples may not always retain the postzygotic mutation after culture cycles[32]. As previously reported[2,3,32], we used in the first place targeted amplicon sequencing in affected skin in all patients to exclude *MTOR* and *RHOA* variants. Patient 2 initially had paired-exome sequencing (coverage of 210X in affected skin vs 80X in blood). We then used *GNA13* targeted amplicon sequencing in affected skin in all patients (coverage ranging from 2374X to 8487X), as well as in peripheral blood, urines and buccal swabs. We amplified regions of interest using custom intronic primers and long-range polymerase chain reactions with the PrimeSTAR GXL DNA Polymerase (Takara Bio, Saint-Germain-en-Laye, France). We pooled, purified, and quantified polymerase chain reaction amplicons from each affected individual. We prepared libraries using the Nextera XT DNA Sample Preparation kit (Illumina, Paris, France). We performed paired-end sequencing reactions of 150-bp reads on a MiSeq platform using 300-cycle reagent kits (Illumina, Paris, France). We assessed the quality of sequencing reads with FastQC and removed sequencing adapters and low-quality bases using Trimmomatic. We aligned reads to the human genome reference sequence GRCh37/hg19 with the Burrows-Wheeler Aligner. We performed realignment around insertions and deletions using the Genome Analysis Toolkit (GATK). Picard and the GATK were used to mark PCR duplicates and collect quality-control, sequencing depth, and coverage metrics.

### Plasmids

pcDNA3.1 and pLVX plasmids encoding for WT, R200K or Q226L Gα$_{13}$ containing an internal YFP tag were previously described[33] and provided by Mikel Garcia-Marcos. The pEGFP-C3-RHOA Q63L construct was reported[34]. Plasmids encoding for Gag, Pol and VSV-G were provided by M. Mangeney (*Institut Cochin*, Paris, France) and T. Henry (*Centre International de Recherche en Infectiologie*, Lyon, France).

### Cells

Melanoma (B16-F0, ATCC # CRL-6322 or SK-MEL-28, ATCC # HTB-72) and keratinocyte (HaCaT, Cell Lines Services, Eppelhein, Germany) cell lines were grown in DMEM (Thermo Fisher Scientific) with 10% foetal calf serum (FCS). Primary Normal Human Epidermal Melanocytes (NHEM) were purchased from Cell Systems (CSC 2HM1) and were grown in complete classic medium with Serum and CultureBoost™ (Cell

Systems − 4Z0-500). For some experiments, NHEM cells were starved with a culture medium without serum (Cell Systems − 4Z3-500).

### Transfection and transduction

Seventy percent confluent B16-F0 were transfected according to the manufacturer's protocol with the lipofectamine 2000 transfection reagent (Thermo Fisher Scientific) and cultured for 48 h including an overnight cell starvation step before the functional assays.

Lentivirus packaging was performed in HEK293T cells by co-transfection of the lentiviral plasmid Gα$_{13}$-pLVX encoding WT or mutant Gα$_{13}$ with the packaging plasmids pVSVG, p8.9 and pREV. The media were changed the next day and collected after 24 h for centrifugation to harvest the lentiviruses.

SK-MEL-28 cells were seeded in flat bottom 96-well plates (Corning, Falcon, $8.10^4$ cells/well) and incubated with the lentiviruses for 2 days until confluence.

Depending on the experiment, $10^5$ or $5.10^5$ NHEM were seeded in 6-well plates (Falcon, #353046) and incubated with lentivirus for 2.5 days in culture medium containing serum and CultureBoost. Cells were then washed with PBS and starved with serum-free culture medium for 12 h.

The percentage of infected cells was analysed by flow cytometry (BD FACSCalibur) at day 3 after infection by analysing YFP expression.

### Quantitative real-time polymerase chain reaction

SK-MEL-28 cells were transduced with Gα$_{13}$ lentiviral constructs as described above. The percentages of cells expressing the relevant Gα$_{13}$ constructs were the following: WT (50 +/− 4 %), R200K (47 +/−6 %), Q226L (48 +/− 3 %). The whole cell populations were then tested for ARPC5 transcripts expression as described[12]. Briefly, RNA was extracted with the kit (QIAGEN, 74104) according to the manufacturer instruction. cDNA was generated using the Maxima Synthesis kit (ThermoFisher, K1641). The expression level of ARPC5 was analysed using SYBR-Green Master Mix (Bio-Rad, #1725271) and normalised to GAPDH expression. The primers used are the following: GAPDH (Fwd 5′−3′ GTCTCCTCTGACTTCAACAGCG; Rv 5′−3′ AACCGATGTCTTG TCCCACCA), ARPC5 (Fwd 5′−3′ AGAGCCCGTCTGACAATAG; Rv 5′−3′ CAG TCAAGACACGAACAATG).

### Pharmacological treatments

RHOA or ROCK inhibitions were performed in serum-free medium incubation for 4 h and overnight using Rho Inhibitor I (Cytoskeleton Inc., #CT04) or Y27632 (Calbiochem) at final concentrations of 2 μg/ml and 5 μM, respectively.

### Immunocytochemistry

Cells were fixed with 4% paraformaldehyde (Electron Microscopy Sciences) in PBS for 10 min at room temperature (RT) and permeabilized with 0.1% Saponin (Fluka Biochemika), 0.2% BSA (Sigma) for 15 min at RT. The latter buffer (permeabilization buffer) was used as a washing buffer all along the labelling. Cells were incubated with primary antibodies anti-pMLC (Cell Signaling Technology, #3671S, 1/100), anti-vinculin (Life Technologies, #700062, 1/100) or anti-RHOA-GTP (CurieCoreTech, 1/100) for 1 h at RT. For co-culture experiments, the primary antibodies used were anti-TRP1 (Santa Cruz, #G-9, 1/300) to stain melanosomes and anti- wide spectrum cytokeratin (abcam, #ab9377, 1/200) to stain keratinocytes. For YAP and TAZ stainings, cells were permeabilized with 0.1 % Triton X-100 buffer for 10 min and washed twice in permeabilization buffer. Cells were then incubated for 45 min with the primary YAP (Cell Signaling Technology, #14074) or TAZ (abcam, #ab242313) antibodies at RT. Cells were then washed and treated with appropriate secondary Alexa Fluor-conjugated antibodies anti-rabbit, anti-mouse or anti-human (Invitrogen) for 30 min at RT and washed again. To visualise the F-actin in parallel, Alexa Fluor 647-conjugated phalloidin was added along with the secondary antibody

incubation. A Hoechst staining in PBS was applied to the cells for 5 min at RT in darkness. Coverslips were finally washed with PBS and mounted with Fluorsave (Calbiochem) overnight to preserve the fluorescence. Image acquisitions were performed with a 100× oil immersion objective with a wide-field Nikon TE2000 microscope equipped with a CMOS (ORCA-flash4.0 LT, Hamamatsu) or Micro-MAX:1300Y (Princeton) camera. Images were acquired with the Metafluor software. Imaging analysis was performed using the ImageJ software, where the fluorescence intensity was measured as a mean grey value. The perimeter, the solidity and the circularity were calculated by the shape descriptors of the software. The degrees of YAP and TAZ nuclear translocation was quantified on Fiji (Fiji Is Just ImageJ) from randomly-acquired images by measuring the mean fluorescence intensity of the staining in the nucleus divided by the mean fluorescence intensity in the cytosol using the same size region. A ratio higher than 1 was considered to be the hallmark of a cell presenting YAP or TAZ nuclear translocation.

## Biochemistry
Protein extraction for whole cell lysate analysis in B16-F0, SK-MEL-28 and NHEM cells was performed in RIPA buffer, supplemented with 1% protease inhibitor (Roche, #48679800) and DNAse Benzonase (Merck, #E10145K). 50 μg of protein lysate was resolved on 10% Tris/Glycin SDS-Polyacrylamide gels. Proteins were subsequently transferred to a polyvinylidene difluoride (PVDF) membrane (Fisher Scientific, #88518). Membranes were blocked for 1 h with 5% milk solution (diluted in TBS-0.1% TWEEN 20) and afterwards incubated overnight with primary antibodies diluted in the TBS-0.1% TWEEN 20 buffer containing 5% BSA (Sigma, #A7030-500). The antibodies used were anti- YAP/TAZ (D24E4) (Cell Signaling Technology, # 8418, 1/1000) which detects both YAP and TAZ proteins, anti- Gα₁₃ (abcam, # ab128900, 1/1000) and α-tubulin (Sigma, #T9026) as a loading control. Primary antibodies were revealed with a secondary HRP-coupled anti-rabbit antibody (Jackson immunoResearch, #111-035-144), diluted in TBS- 0.1% TWEEN 20, for 1 h at RT. Bands were detected using Super-Signal West Pico Plus chemiluminescence (Thermo Fisher, #34580) and the Fusion FX imaging system.

## Annexin V staining
Cells were harvested with Trypsin/EDTA three days after transfection, washed with PBS, and resuspended in binding buffer containing the fluorochrome-conjugated Annexin V. After incubation for 15 min away from light, cell suspension was diluted with the binding buffer, centrifuged to pellet the cells, and analysed by flow cytometry.

## Proliferation assay
Cells were seeded and transfected with lipofectamine or transduced with lentiviruses the next day. 24 h later, the cells were stained with 5 mM CellTrace violet (CTV) Cell Proliferation kit (Invitrogen, #C34571) diluted in PBS for 20 min at RT. Subsequently, the cells were washed twice: once with culture medium, and then with FCS. The cells were grown in DMEM and analysed by FACS Fortessa (BD Biosciences) for CTV fluorescence intensity at days 1, 2 and 3 following the staining. Live cells were identified (FSC-A vs SSC-A) and doublets were excluded (SSC-A vs SSC-H and FSC-A vs FSC-H). Transfected YFP⁺ cells were identified starting from non-transfected YFP⁻ cells. CTV fluorescence intensity was then quantified at different days on YFP⁺ cells using the Violet Laser.

## Melanocyte-keratinocyte co-culture
HaCaT cells were seeded at $2.5 \times 10^5$ cells/well in six-well plates. The next day, B16-F0 cells were added to each well containing the keratinocytes at a keratinocyte to melanocyte seeding ratio of 2.5: 1. Following transfection, a serum-free medium containing 100 nM MSH was added and left for two days. Cultures were then fixed and labelled for immunofluorescence microscopy analysis. Upon random acquisition of microscopy fields, the number of melanosomes and keratinocytes was counted.

## Melanin quantification
100 nM Melanin Stimulating Hormone (MSH, Merck) was added in serum-free medium for 48 h following transfection. For the melanin content assay, cells were harvested with Trypsin/EDTA and the amount of melanin was quantified as followed: A comparable number of cells were lysed with 100 μl 1 N NaOH, 10 % DMSO, heated at 80 °C for 1 h 30, and vortexed repeatedly to homogenise. Cell extracts were placed in 96-well plates in duplicate. The relative melanin content was determined by measuring absorbance at 490 nm with a Clariostar reader, and a wide range of pure synthetic melanin (Merck).

## Scratch wound assay
SK-MEL-28 cells were transduced with Gα₁₃ lentiviral constructs as described above. The percentages of cells expressing the relevant Gα₁₃ constructs were around 82 +/− 5%. The Essen Bioscience WoundMaker was used to create scratch-wounds of a standardised width (~600 μm) on cell monolayers. Cells were tracked using the automated live-cell Essen IncuCyte Zoom live-cell microscopy system, taking an image every two hours. The Incucyte S3 software was used for image analysis and wound confluence determination.

## Statistics
One-way ANOVA or Friedman tests were performed with GraphPad (Prism). *$P < 0.05$; **$P < 0.01$; ***$P < 0.001$; ****$P < 0.0001$. All graphs shown are representative of at least 3 independent experiments.

## Reporting summary
Further information on research design is available in the Nature Portfolio Reporting Summary linked to this article.

## Data availability
The authors confirm that all relevant data are included in the paper and/or its Supplementary Information files. Source data are provided with this paper.

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

## Acknowledgements

We sincerely thank the patients and their families for participating in this study. We thank Mikel Garcia-Marcos, Marianne Mangeney and Thomas Henry for providing plasmids, Graça Raposo (Curie Institute, Paris) for discussions and comments on the manuscript, Thomas Guilbert (IMAG'IC Facility) and the CYBIO facility for technical assistance. This study was funded by *Agence Nationale de la Recherche* (ANR; RIDES 2019, Cytoskinflam 2023), Inserm, Inserm Transfert (COPOC 2023), and *Université Paris Cité* (UPC). R.E.M. was supported by a postdoctoral fellowship from ANR. A.I. was supported by an international PhD fellowship from UPC, a European Society for Immunodeficiencies fellowship and by Fondation pour la Recherche Médicale (FRM Grant number FDT202404018275).

## Author contributions

J.D. and P.V. designed the project and acquired funds. M.V., S.B., F.M.-P., F.B., N.O. and J. M.-H. are clinicians who monitor the described patients. A.S., P.K., Y.D., L.F. and P.V. performed the genetic characterization of the patients. A.S., P.K. and P.V. compiled the clinical data. R.E.M., A.I and R.T. performed the experiments under the supervision of J.D. R.E.M. and J.D. wrote the manuscript, with contributions from P.V., A.I., P.K., A.S., S.B., F.M.-P. and J.M.-H.

## Competing interests

The authors declare no competing interests.

## Additional information

[1]Université Paris Cité, CNRS, Inserm, Institut Cochin, Paris, France. [2]UFR des Sciences de Santé, Inserm - Université de Bourgogne UMR1231 GAD "Génétique des Anomalies du Développement", FHU-TRANSLAD, Dijon, France. [3]Oncobiologie Génétique Bioinformatique, PCBio, Centre Hospitalier Universitaire de Besançon, Besançon, France. [4]Unité de Génétique clinique, Service de génétique médicale, CHU de Nantes - Hôpital Mère-Enfant, Nantes, France. [5]Nantes Université, Department of Dermatology, CHU Nantes, INRAE, UMR 1280, PhAN, Nantes, France. [6]MAGEC Reference Centre for Rare Genetic Skin Diseases, Paediatric Dermatology Unit, Department of Dermatology, CHU de Bordeaux - GH Pellegrin, Bordeaux, France. [7]Service de dermatologie, CHU de Toulouse - Hôpital Larrey, Toulouse, France. [8]CHU Dijon, Unité Fonctionnelle "Innovation diagnostique dans les maladies rares", FHU-TRANSLAD & Institut GIMI, Dijon, France. [9]CHU Dijon, Centre de Génétique et Centres de référence Anomalies du Développement et Déficience Intellectuelle, FHU-TRANSLAD & Institut GIMI, Dijon, France. [10]MAGEC Reference Centre for Rare Genetic Skin Diseases and Paediatric Dermatology Unit, Department of Paediatrics, University Hospital Dijon-Bourgogne, FHU-TRANSLAD & Institut GIMI, Dijon, France. [11]Rare Disease Collaborative Network (RDCN) Adult Mosaic Disorders Clinic, St John's Institute of Dermatology, Guy's and St Thomas' NHS Foundation Trust, London, United Kingdom. [12]Present address: Department of Cell Physiology & Metabolism, Faculty of Medicine, University of Geneva, Geneva, Switzerland. [13]Present address: National Center of Genetics (NCG), Laboratoire national de santé (LNS), 1 Rue Louis Rech, Dudelange, Luxembourg. [14]These authors contributed equally: Pierre Vabres, Jérôme Delon. [15]Deceased: Franck Boralevi.
✉e-mail: vabres@u-bourgogne.fr; jerome.delon@inserm.fr

