## [Transparent Peer Review file · Nature Communications]

A postzygotic GNA13 variant upregulates the RHOA/ROCK pathway and alters melanocyte function in a mosaic skin hypopigmentation syndrome

Corresponding Author: Dr Jerome Delon

Version 0:

Reviewer comments:

Reviewer #1

(Remarks to the Author)

This is a report of a recurrent postzygotic variant in GNA13 in 4 patients with Hypomelanosis of Ito and a similar phenotype. The studies reported here suggest that this mosaic variant interferes, via hyperactivation of the RHOA/ROCK signalling pathway, with transfer of melanin from melanosomes to keratinocytes and possibly also with migration of melanocytes from the neural crest, resulting in the cutaneous phenotype of white lesions in a Blaschkoid distribution.

I would encourage authors to consider the following points to strengthen this paper.

1. The diagnostic term used throughout the paper is Hypomelanosis of Ito while in the abstract the patients are referred to as exhibiting pigmentary mosaicism. Pigmentary mosaicism is the preferred term, because it is more helpful to lump both the hyper and hypopigmented forms together, not least because in some patients both co-exist. Also, there is a general move towards abandoning eponyms. I can understand that the authors wish to emphasise that hypopigmentation is associated with this variant so a possible compromise would be to refer to it as Hypopigmented pigmentary mosaicism.
2. A diagram of the relevant components of the RHOA/ROCK pathway would be helpful in explaining the functional experiments.
3. Page 7 lines 179-187: the phenotypes could be described in more detail in the text. Even in supplementary table 1 there is a lot of missing data including age of patient 2 and nature of the hearing defect in 2 patients.
4. "These patients also have wound healing issues, teeth and ocular anomalies, gastroenterological and nephrological defects suggests more phenotypic overlap than can be found among the limited details given in supplemental table 1. The teeth and ocular anomalies are quite different, and only 1/4 had renal anomalies. The GI similarities are however striking (atresias in 3/4 patients). Regarding wound healing, Only the biopsy wounds are mentioned and only in two patients was there a general issue with wound healing and was it confined to affected skin?"
5. Page 11 lines 316-317: This sentence does not make sense: "We have found evidence leading to hypomelanosis of Ito."
6. P11 lines 325-327: as with other mosaic conditions, diagnosis of mutation testing in G13-related mosaic disorders should be performed on a biopsy from affected skin. This is not necessarily the case. If deep sequencing is available, testing for mosaic conditions can be successfully performed on blood, only proceeding to skin biopsy if negative.
7. P11 lines 339-40: the authors imply that the current paradigm regarding the cause of Hypomelanosis of Ito relates just to melanin production. However, Taibjee et al previously concluded that the large number of chromosomal locations reported in hypomelanosis of Ito and pigmentary mosaicism reflected the many different ways in which pigmentation could be disturbed (Taibjee SM et al Br J Dermatol.2004;151(2):269-82).
8. Page 12 line 354: given the molecular links with McCune Albright syndrome, can this be reconciled with the Blaschkoid hyperpigmentation in that disorder?
9. Figure 1 (f): stellate pigmentation cannot be discerned.

Reviewer #2

(Remarks to the Author)

In this manuscript the authors describe 4 unrelated individuals with the same recurrent variant in GNA13, a G protein

component. Each of these individuals possesses a cutaneous abnormality called hypomelanosis of ito, with associated developmental malformations. The presence of 4 unrelated individuals with the same, postzygotic variant and the same rare phenotype is unequivocal proof for this gene-disease association. There is also a great deal of work, using a variety of in vitro cell lines, to better understand the mechanisms by which this GNA13 mutation (p.R200K) acts. The authors show that

A B16-F0 melanoma cell line with R200K and E226L GNA13 overexpression have more filamentous actin and are rounder than cells overexpressing GNA13 without these substitutions

These same cells have increased activated myosin light chain and reduced vinculin expression, consistent with an impact on cytoskeleton and cell adhesion

These same R200K and E226L based effects- namely the increase in pMLC, the increased F-actin and cell roundness, were removed in the presence of RHOA and ROCK pharmacologic inhibition, consistent with these effects downstream of GNA13 being mediated by Rho/ROCK signaling

Using a different cell line due to increased transfection efficiency (SK-MEL28) they show a reduction in cell migration using the scratch assay in the presence of R200K or E226L overexpression

Lastly, using the B16-F0 line again, bc it is responsive to MSH, they show that some cytoskeletal effects of MSH stimulation (number, length of dendritic extensions) are abated by the presence of R200K or E226L overexpression. Additionally, quantification of # melanosomes per co-cultured keratinocyte shows a reduction in the number of transferred melanosomes from R220K or E226L overexpressing melanocytes, overall suggesting that perhaps the clinical phenotype of hypomelanosis is due to local inhibition of melanosome transfer.

Overall the paper is well written, clear, and the conclusions appear well supported. A clear clinical management change- namely testing HI patients for GNA13 mutations- is supported here. However, there is an abundance of cell based mechanistic experiments and discussion but very little towards the more clinical genetic side My thoughts below are an attempt to strengthen this paper further:

- 1) Can the authors provide a bit more information about the affected skin biopsy samples. Were they taken from the regions of streaky hypopigmentation shown in figure 1? I do not see any scars or old biopsy sites in these pictures. Or, do the authors mean they were just taken from an area of alopecia or a limb that was hypoplastic? These details are critical for the diagnosis of other patients.
- 2) Was sequencing performed on DNA from cultured fibroblasts or directly from a punch? The authors note that patient 2 underwent 210X exome sequencing, but in the other three patients "targeted ultra-deep sequencing was performed. Please provide details about this ultra-deep sequencing (what genes included, what coverage, etc)
- 3) The authors suggest in the discussion that performing experiments on directly cultured cells from these patients is limited by ethical considerations due to their poor wound healing. However they also state that diagnosis here is only possible through a biopsy. Can they please describe what complications occurred for these patients, all of whom did indeed receive a skin biopsy? Is this truly a clinical concern, or just a justification for not doing these experiments on patient derived cells?
- 4) At this level of mosaicism (21-36%), I suspect this variant is present in many other tissues, and perhaps is just absent from the blood cells. The authors need to test some other samples from these individuals (besides blood, which they showed) to determine if the GNA13 mutation is detectable in other tissues. This is very important as it could increase the number of potential biopsy sites for clinicians, and given that the phenotype is widespread in the body, the variants absence from blood does not mean it is absent in other tissues.
- 5) In line with the point above: if possible, the authors should test cfDNA derived from these individuals blood samples to determine if the mutation is present in plasma derived cfDNA.
- 6) There needs to be some discussion or speculation about the widespread and diverse aspects of the "hypomelanosis of ito phenotype. In the clinical table, these individuals are listed as having a variety of major malformations such as parietal alopecia, jejunal atresia, acral anomalies, hemihypoplasia, and coloboma of the superior eyelid. Many of these features overlap with other mosaic conditions, notably encephalocraniocutaneous lipomatosis.
- 7) The authors have a line (page 12, top of page) that the mosaicism we describe here may protect patients from cancer progression. This line is misleading to me or I am not understanding. I think they are trying to explain the absence of cancer in their patients despite the fact that R200K is a cancer associated mutation. But there is a great deal of precedent in the mosaic conditions for the presence of oncogenic mutations and the absence of cancer. For example, PIK3CA related overgrowth patients rarely develop cancer for reasons that remain unclear. The authors have not tested enough cellular compartments in their patients to know where the mutation is present or absent, other than the blood. Also three of their patients remain quite young (one has no age listed), so they cannot yet be certain these individuals may not go on to develop cancer. Can the authors please clarify or change this line?
- 8) I do appreciate the attempt to discuss cancer risk in HI patients, and the authors should probably refer to previous publications where this has been described. The finding of an oncogenic mutation in HI patients certainly might be an explanation for some of the previous reports of cancer and HI association, and I think the authors should note this, but also note that the cancer risk in HI relatively lower than what one might expect, for reasons that remain mostly unknown.
- 9) What is the developmental expression of GNA13 in humans and would it be possible that some isolated birth defects such as jejunal atresia could be caused by mosaic GNA13 mutations, even in the absence of hypomelanosis?
- 10) My last comment (and only comment on the cell based functional side of the paper) is that many of the assays (f-actin

amount, cell roundness, vinculin expression) have a large amount of variance and a great deal of overlap with the “WT” conditions, despite the significant p-values. Can the authors comment on this? I am guessing it is likely due to variance in GNA13 transfection rates- if so is it possible to correct for this? I am not an expert on in vitro cell based experiments (hence all my comments on clinical aspects), so if this reflects normal variance that is fine, but perhaps a sentence in the discussion would be helpful?

Reviewer #3

(Remarks to the Author)

This is an interesting paper reporting on GNA13 activating mutations as an explanation for a subset of patients with the clinical pattern of mosaicism termed Hypomelanosis of Ito.

The authors demonstrate this in 4 patients using exome sequencing in one and targeted sequencing of GNA13 in 3 patients. The R200K mutation has been studied previously in bladder cancer cells and shown to be a hotspot that perturbs YAP-TAZ signalling via Rho-GEF/GTPase. There have been some cellular contexts where GNA13 mutations lead to inactivation rather than activation, and hence the characterisation in cells with a melanocyte lineage are important (10.1074/jbc.AC120.014698).

The majority of the paper investigates the consequence of this mutation in cellular models, principally melanoma cell lines. Key readouts are changes in cellular morphology and expression of proteins visualised and quantified on ICC cells. Comparisons are made between the R200K mutation and a more extensively studied change at codon 226.

The basis for hypopigmentation posited ultimately is the reduction in melanosome transfer, or alternatively reduced numbers in the skin due to impaired migration to the skin.

Points to consider:

Investigation on the effect of the GNA13 R200K change on cell proliferation and YAP TAZ signalling are lacking and would give insights in this cellular context.

- 1) What are the basal levels of YAP and TAZ in the cell lines used (SKMEL28 and B16F0)? This can vary and can impact on cytoskeleton remodelling and cell migration – see <https://onlinelibrary.wiley.com/doi/10.1111/pcmr.13013>
- 2) It would be informative to know the consequence on YAP TAZ signalling when GNA13 R200K are transfected in the cell lines used, as it has been shown that YAP levels can influence the number of focal adhesions in melanoma cells.
- 3) Do these GNA13 R200k transduced cells proliferate more?
- 4) Have these changes been explored outside a cancer cell line context such as in NHEMs? This may give a closer representation to what happens in embryonic skin development.

Minor points;

The bioinformatic pipeline used was not clear have other genes been excluded in the targeted sequencing of these patients? What variant caller was used, and was deduplication of PCR reads used? The % of reads (21-36%) was higher than expected when compared to other mosaic skin conditions affecting pigment cells (eg <5%).

Version 1:

Reviewer comments:

Reviewer #1

(Remarks to the Author)

Thank you for addressing the points raised previously. The following remain:
Page 12, paragraph beginning line 304: The authors justify the use of melanocytes derived from skin biopsies and maintain that diagnosis of mutation testing in G13-related mosaic disorders should be primarily performed on a biopsy from affected skin. There should be an acknowledgement here that other, less invasive sources of DNA might be used, in accordance with their response to reviewers stating we thank the Reviewer for suggesting we test the presence of the GNA13 mutation in other tissues, as it turns out that buccal epithelium could be a less invasive source of cells for some of these patients. In clinical practice, there is usually a wider differential diagnosis than G13-related mosaic disorders, and (particularly in children, and since wound-healing might be impaired) an initial less invasive test, such as venesection or buccal smear, is preferable, usually covering a range of mosaic skin disorders, some of which can be detected on deep sequencing in blood. If that is negative it is reasonable to proceed to skin biopsy and perhaps more targeted testing.
Page 7, line 156: either both mutants should presumably be either or both mutants.

Page 9, line: exhibit similar features than should be exhibit features similar to.

Page 12, line 298: what is meant by Defects in cell 6?

Page 12, line 302: the word Consistently means invariably not consistent with that. I suggest changing the wording to Consistent with that hypothesis

Page 13, line 323: The mechanisms responsible..remains should be The mechanisms responsible..remain

Page 20, line 548: the numberwere countedshould be the numberwas counted

Reviewer #2

(Remarks to the Author)

Thanks to the authors for addressing all of my concerns and comments. The manuscript is significantly improved, and the results important and exciting to our field. I have no additional comments.

Reviewer #3

(Remarks to the Author)

Thank you for doing the additional experiments on normal melanocyte cell lines. This adds to the work and the YAP nuclear localisation seen is also helpful. Overall, I am satisfied with the experiments carried out and I think a strong case is made for the impact of this pathogenic variant in GNA13 in these patients affecting the function of melanocyte cell lines.

It is pleasing that there is some alignment of the outcomes of the melanoma cell lines and the melanocyte cell line for readouts studied, however this must be cautiously interpreted (response to point 4) and some additional clarification is needed:

For Figure 3, it could be argued that the impact of the R200K mutation on circularity and f-actin content is more pronounced in the NHEM cells. Could this be shown in Figure 3 as a "side by side" comparison rather than in Supplementary Figure 3? This could align well with the narrative for the paper that the effects seen on cytoskeleton remodelling are similar in both normal and cancer cell lines as the authors have done in the revised text line 161 page 7.

For Figure 6 a, it is good that the western blot of presumed basal levels of YAP and TAZ are indicated. Please show the expression levels using immunoblotting of YAP and TAZ both in cells with GNA13 WT overexpression as well as R200K and Q226L across the 3 cell lines.

For Figure 6b, the claim of a lack of change in proliferation in the abstract following GNA13 R200K overexpression holds for a melanoma cell line (B16-F0) however this may be different in other melanoma cell lines (SKMEL28) and particularly normal melanocytes (NHEM). It would be important to show this across 3 cell lines, given that in a number of cancers, YAP hyperactivation leads to cellular proliferation (<https://doi.org/10.1038/s41467-024-46531-1>). In the event of variation across cell types, the interpretation of NHEM cells should be favoured, as the context in the paper is that of melanocytes in human skin development.

Minor points:

I suggest that given the different types of cell lines now used in the paper, that this is indicated within the figures for the ease of the reader to follow the narrative.

For the QPCR data Supp Fig 7c the gene should be the title of the graph.

Dear Reviewers,

We thank you for your appreciation of the article and constructive comments, which have greatly helped us to improve the manuscript. In the revised version, we have been able to answer all your concerns and recommendations. Changes from the previous paper are now highlighted in yellow in the revised version of the article.

Briefly, additional information includes:

- Clinical symptoms of each patient have been more precisely indicated in Table 1 that we have now included in the main text.
- Additional patients' pictures are provided (Fig. 1).
- The presence of the *GNA13* mutation has been tested in additional patients' samples (blood, urines and buccal swabs) (Table 1).
- The use of primary Normal Human Epithelial Melanocytes (NHEM) has shown that the $G\alpha_{13}$ R200K and Q226L mutants also elicit an increase in actin filaments polymerization, circularity, solidity, and a decrease in cell perimeter (Sup. Fig. 3a). P-MLC (Sup. Fig. 3b) and RHOA-GTP (Sup. Fig. 3c) levels are also increased by the two studied $G\alpha_{13}$ R200K and Q226L mutants in NHEM. These results fully confirm our data shown in the B16-F0 and SK-MEL-28 melanocytic cell lines (Fig. 3, Fig. 4a-b, Fig. 5a,d-g and Sup. Fig. 2a-e).
- YAP/TAZ expression (Fig. 6a), nuclear translocation (Fig. 6b-c, Sup. Fig. 6 and Sup. Fig. 7a-b) and transcriptional activity (Sup. Fig. 7c) were studied in B16-F0, SK-MEL-28 and NHEM cells expressing WT, R200K or Q226L $G\alpha_{13}$.
- The effects of $G\alpha_{13}$ R200K and Q226L mutants on cell proliferation are shown in Fig. 6d.
- A simplified diagram of the RHOA/ROCK pathway which includes the majority of our findings has been added (Sup. Fig. 9).

Other additional data and more detailed information are also provided in the point-by-point response below.

REVIEWERS' COMMENTS

Reviewer #1 (Remarks to the Author):

This is a report of a recurrent postzygotic variant in GNA13 in 4 patients with Hypomelanosis of Ito and a similar phenotype. The studies reported here suggest that this mosaic variant interferes, via hyperactivation of the RHOA/ROCK signalling pathway, with transfer of melanin from melanosomes to keratinocytes and possibly also with migration of melanocytes from the neural crest, resulting in the cutaneous phenotype of white lesions in a Blaschkoid distribution.

I would encourage authors to consider the following points to strengthen this paper.

1. The diagnostic term used throughout the paper is Hypomelanosis of Ito while in the abstract the patients are referred to as exhibiting pigmentary mosaicism. Pigmentary mosaicism is the preferred term, because it is more helpful to “lump” both the hyper and hypopigmented forms together, not least because in some patients both co-exist. Also, there is a general move towards abandoning eponyms. I can understand that the authors wish to emphasise that hypopigmentation is associated with this variant so a possible compromise would be to refer to it as “Hypopigmented pigmentary mosaicism”.

The best denomination for linear depigmentation along Blaschko's lines is indeed still a matter of debate, and no consensus exists. We tend to stick to the name “hypomelanosis of Ito”, as it has long been used, particularly in our previous papers on *RHOA* and *MTOR* mosaicism. We consider this term as a description, not a diagnosis, in keeping with V. Sybert's paper which is cited in the *References* section (Reference # 1). However, we agree with the reviewer that the term “pigmentary mosaicism” has also been increasingly used. Hence, we chose to keep both denominations, which are now both mentioned and clarified in the introduction section (page 4, lines 71-75). Ultimately, only a common genetic definition will allow to classify these mosaic conditions beyond their mere clinical description, which has shown its limitations, particularly in mosaic syndromes due to highly variable expression (page 4, lines 77-81). Thus, we have added a reference to the “dyadic” definition of these genetic conditions (Reference # 4: Biesecker *et al.*, 2021), which has allowed us to understand other mosaic syndromes (Klippel-Trenaunay syndrome, Proteus syndrome, and phakomatosis pigmentovascularis).

2. A diagram of the relevant components of the RHOA/ROCK pathway would be helpful in explaining the functional experiments.

We completely agree with the reviewer, and we have added a simplified diagram of the RHOA/ROCK pathway which includes the majority of our findings (Supplementary Fig. 9) (page 12, lines 318-319).

Supplementary Fig. 9:

3. Page 7 lines 179-187: the phenotypes could be described in more detail in the text. Even in supplementary table 1 there is a lot of missing data including age of patient 2 and nature of the hearing defect in 2 patients.

We agree that a better clinical description of the phenotype was needed, and we have now strengthened this chapter in the manuscript. Overall, all four patients share clinical similarities with patients with RHOA-related neuroectodermal syndromes (facial and acral anomalies), which we have added in the *Introduction* (page 4, lines 90-92). We have also recently gathered additional clinical information whenever possible, and we have now designed a more accurate and complete clinical description in the main text and in Table 1 that we have now put in the main text too (page 6, lines 109-126).

The age of Patient 2 had indeed been omitted in Table 1. We apologise for that. We have now added this information.

Regarding hearing defect, in patient 2, it appears to be both a transmission (otosclerosis) and perception (cochlear nerve agenesis) defect. In patient 3, we have insufficient data yet, as the patient is a toddler with chronic otitis media.

4. “These patients also have wound healing issues, teeth and ocular anomalies, gastroenterological and nephrological defects” suggests more phenotypic overlap than can be found among the limited details given in supplemental table 1. The teeth and ocular anomalies are quite different, and only 1/4 had renal anomalies. The GI similarities are however striking (atresias in 3/4 patients). Regarding wound healing, Only the biopsy wounds are mentioned and only in two patients – was there a general issue with wound healing and was it confined to affected skin?

Gastrointestinal tract atresia appears indeed to be a non random manifestation in this mosaic condition, as it was found in 3 patients (see page 6 lines 120-122 and page 12 lines 297-304).

Two patients experienced delayed wound healing for several months after skin biopsy performed on affected skin for genetic testing (see modifications in the manuscript page 6 lines 122-123, page 12 lines 297-308, Table 1 and Fig. 1a). Despite the small patient sample size, we consider this wound healing problem as a relevant clinical feature of the *GNA13* mosaic syndrome. It is supported by the role of the RHOA/ROCK pathway in cell migration (page 15, lines 391-392) and the role of YAP in dermal regeneration (page 14, lines 353-358).

We agree that tooth anomalies may not be specific. Ocular involvement consisted of coloboma of the iris or eyelid, and may be a relevant manifestation of the syndrome (page 6, lines 123-124). Hydronephrosis and megaureter was indeed found in one patient only. Thus, we have modified the manuscript accordingly.

5. Page 11 lines 316-317: This sentence does not make sense: “We have found evidence leading to hypomelanosis of Ito”.

This sentence has been removed. It was a remnant from a previous draft. We thank the Reviewer for pointing this out.

6. P11 lines 325-327: “as with other mosaic conditions, diagnosis of mutation testing in *Gα13*-related mosaic disorders should be performed on a biopsy from affected skin.” This is not necessarily the case. If deep sequencing is available, testing for mosaic conditions can be successfully performed on blood, only proceeding to skin biopsy if negative.

In our opinion, skin biopsy remains the first step for genetic testing. We performed ultradeep sequencing on blood DNA (2000x to 3000x) without finding the variant in all four patients (Table 1) (page 6, line 135). However, we did not perform analysis of circulating plasma cell free DNA, as we do not routinely use this technique. Our experience with other disorders of somatic mosaicism (related to *PIK3CA*, *RHOA* or *MTOR*) is that the postzygotic variant is unlikely to be found on the blood, which should not be used as the first step for genetic diagnosis.

7. P11 lines 339-40: the authors imply that the current paradigm regarding the cause of Hypomelanosis of Ito relates just to melanin production. However, Taibjee et al previously concluded that the large number of chromosomal locations reported in hypomelanosis of Ito and pigmentary mosaicism reflected the many different ways in which pigmentation could be disturbed (Taibjee SM et al Br J Dermatol. 2004;151(2):269-82).

We agree that various defects in the pigmentation process may explain pigmentary mosaicism. Indeed, we previously found increased melanin production in *KITLG*-related naevoid hypermelanosis (PMID: 28257793), as well as a decrease in intra-keratinocytic melanosomes and a defect in maturation of melanosomes in *MTOR*-related hypomelanosis of Ito (Reference # 2: PMID: 33833411). Here, we show a

defect in the transfer of melanosomes to keratinocytes, as well as a defect in melanocytes migration. Thus, we have modified this part accordingly (page 13, lines 324-333) and quoted the Taibjee et al. paper (Reference # 14).

8. Page 12 line 354: given the molecular links with McCune Albright syndrome, can this be reconciled with the Blaschkoid hyperpigmentation in that disorder?

G protein α -subunits are indeed involved in several mosaic pigmentation disorders: *GNAS* in McCune Albright syndrome, or *GNA11* in extensive dermal melanocytosis/phakomatosis pigmentovascularis. However, all have pigmentation patterns that are usually different from Blaschko's lines. Unfortunately, we still have no explanations to offer to the pattern of pigmentation based on the gene involved (page 14, lines 370-378).

9. Figure 1 (f): stellate pigmentation cannot be discerned.

No stellate pigmentation is indeed visible. Thank you for pointing out this error. We have deleted this and have provided a more accurate description of the pigment pattern in Table 1 and in the *Results* section of the manuscript (page 6, lines 112-118). Also, we have modified Figure 1 to include pictures more accurately showing the pigmentation pattern, including at the biopsy site (Fig. 1a).

Reviewer #2 (Remarks to the Author):

In this manuscript the authors describe 4 unrelated individuals with the same recurrent variant in GNA13, a G protein component. Each of these individuals possesses a cutaneous abnormality called “hypomelanosis of ito”, with associated developmental malformations. The presence of 4 unrelated individuals with the same, postzygotic variant and the same rare phenotype is unequivocal proof for this gene-disease association. There is also a great deal of work, using a variety of in vitro cell lines, to better understand the mechanisms by which this GNA13 mutation (p.R200K) acts. The authors show that

- A B16-F0 melanoma cell line with R200K and E226L GNA13 overexpression have more filamentous actin and are rounder than cells overexpressing GNA13 without these substitutions
- These same cells have increased activated myosin light chain and reduced vinculin expression, consistent with an impact on cytoskeleton and cell adhesion
- These same R200K and E226L based effects- namely the increase in pMLC, the increased F-actin and cell roundness, were removed in the presence of RHOA and ROCK pharmacologic inhibition, consistent with these effects downstream of GNA13 being mediated by Rho/ROCK signaling
- Using a different cell line due to increased transfection efficiency (SK-MEL28) they show a reduction in cell migration using the scratch assay in the presence of R200K or E226L overexpression
- Lastly, using the B16-F0 line again, bc it is responsive to MSH, they show that some cytoskeletal effects of MSH stimulation (number, length of dendritic extensions) are abated by the presence of R200K or E226L overexpression. Additionally, quantification of # melanosomes per co-cultured keratinocyte shows a reduction in the number of transferred melanosomes from R220K or E226L overexpressing melanocytes, overall suggesting that perhaps the clinical phenotype of hypomelanosis is due to local inhibition of melanosome transfer.

Overall the paper is well written, clear, and the conclusions appear well supported. A clear clinical management change- namely testing HI patients for GNA13 mutations- is supported here. However, there is an abundance of cell based mechanistic experiments and discussion but very little towards the more clinical genetic side My thoughts below are an attempt to strengthen this paper further:

1) Can the authors provide a bit more information about the “affected” skin biopsy samples. Were they taken from the regions of streaky hypopigmentation shown in figure 1? I do not see any scars or old biopsy sites in these pictures. Or, do the authors mean they were just taken from an area of alopecia or a limb that was hypoplastic? These details are critical for the diagnosis of other patients.

All skin biopsies were indeed taken from hypopigmented skin (page 6, lines 127-133). None of the patients were biopsied on normal skin or patches of alopecia. We have added a picture from Patient 1 in the revised version of Figure 1, where both the pattern of hypopigmentation and the biopsy scar are visible (Fig. 1a).

2) Was sequencing performed on DNA from cultured fibroblasts or directly from a punch? The authors note that patient 2 underwent 210X exome sequencing, but in the other three patients “targeted ultra-deep sequencing” was performed. Please provide details about this ultra-deep sequencing (what genes included, what coverage, etc) Sequencing was performed directly from a punch biopsy (page 6, lines 127-133). Indeed, we previously showed that sequencing on cultured samples may miss the postzygotic mutation (Kuentz et al. 2017 - PMID: 28151489) as the variant may be lost in the cell culture process. We performed targeted amplicon sequencing in affected skin in all patients to exclude *MTOR* and *RHOA* variants, as previously reported (Kuentz et al. 2017 - PMID: 28151489; Vabres et al. 2019 - PMID: 31570889; Carnignac et al. 2021 - PMID: 33833411). Patient 2 had paired-exome sequencing (coverage of 210x in affected skin and 80x in blood), which detected the *GNA13* variant in affected skin (absent in blood). We then performed *GNA13* targeted amplicon sequencing in affected skin in all patients (ultra-deep coverage, ranging from 2374x to 8487x) because of their similar phenotype. We have now added further technical data in the *Methods* (page 16, lines 415-435) and *Results* (Page 6, lines 127-135) sections of the revised manuscript.

3) The authors suggest in the discussion that performing experiments on directly cultured cells from these patients is “limited by ethical ... considerations” due to their poor wound healing. However they also state that diagnosis here is only possible through a biopsy. Can they please describe what complications occurred for these patients, all of whom did indeed receive a skin biopsy? Is this truly a clinical concern, or just a justification for not doing these experiments on patient derived cells?

Two patients experienced delayed wound healing for several months after skin biopsy for genetic testing (see Table 1, new Fig. 1a and modifications in the manuscript: page 6, lines 122-123). One of them (Patient 1) understandably refused further skin biopsy. Despite the small patient sample size, we consider this wound healing issue as a relevant clinical feature of the *GNA13* mosaic syndrome. It is supported by the role of the *RHOA/ROCK* and *YAP* pathways in cell migration and dermal regeneration. This has been highlighted in the manuscript (page 12, lines 298-299; page 13-14, lines 353-358). Despite this possible complication, skin biopsy remains the method required for genetic diagnosis (see below - response to comments 4 & 5).

4) At this level of mosaicism (21-36%), I suspect this variant is present in many other tissues, and perhaps is just absent from the blood cells. The authors need to test some other samples from these individuals (besides blood, which they showed) to determine if the *GNA13* mutation is detectable in other tissues. This is very important as it could increase the number of potential biopsy sites for clinicians, and given that the

phenotype is widespread in the body, the variant's absence from blood does not mean it is absent in other tissues.

The variant allele fraction (VAF), which is indeed quite high, only reflects the level of mutant cells in a given sample (here the skin), not in the whole body. Indeed, we and others have reported highly variable VAFs in different tissues in a patient on *post-mortem* findings, be it in Proteus syndrome (PMID: 27112325) or MTOR-related hypomelanosis of Ito (PMID: 38379111).

In keeping with the reviewer's suggestion, we were able to collect buccal swabs and urine from all four patients, and blood again from three patients (Patient 3 is only two years old and the consent for a blood draw was not obtained). Sequencing depth between 1000x and 3000x was achieved. In Patients 2 and 3, the variant was identified in buccal swabs, with a VAF of 10.6 % and 3.9 %, respectively (see Table below and in the revised manuscript: Table 1 and page 6, lines 133-135). A VAF close to 0 % (or lower than 1 %, which we consider the threshold between a relevant variant and background noise) was found in buccal swabs from the other two patients and in urines from three patients. Unfortunately, despite two trials, we were unable to obtain the results of the sequencing on the urine of Patient 2 on time due to a technical problem. All these data are now included in a new version of Table 1 in the main text of the revised manuscript. No biopsy material from other tissue was available in any of the patients. Thus, although the skin remains one of the most accessible tissues, we thank the Reviewer for suggesting we test the presence of the *GNA13* mutation in other tissues, as it turns out that buccal epithelium could be a less invasive source of cells for some of these patients.

Tissue	Patient 1			Patient 2			Patient 3		Patient 4		
	Buccal swab	Blood	Urine	Buccal swab	Blood	Urine	Buccal swab	Urine	Buccal swab	Blood	Urine
Alternative allele reads number	2	6	0	302	1	NA - failure	76	2	9	0	0
Total reads number	1833	2072	1742	2843	2879	NA - failure	1927	3011	1793	1628	1088
Variant allele fraction	0,1%	0,3%	0,0%	10,6%	0,0%	NA - failure	3,9%	0,1%	0,5%	0,0%	0,0%

5) In line with the point above: if possible, the authors should test cfDNA derived from these individuals blood samples to determine if the mutation is present in plasma derived cfDNA.

We do not routinely perform detection of plasma circulating free DNA in our lab, hence, we did not implement this technique in the limited allotted time to resubmit the manuscript. However, we agree that it may allow detection of mosaicism on peripheral blood, which would be less invasive than affected tissue biopsy.

6) There needs to be some discussion or speculation about the widespread and diverse aspects of the "hypomelanosis of ito" phenotype. In the clinical table, these individuals are listed as having a variety of major malformations such as parietal alopecia, jejunal atresia, acral anomalies, hemihypoplasia, and coloboma of the superior eyelid. Many of these features overlap with other mosaic conditions, notably encephalocraniocutaneous lipomatosis.

We have added a more detailed clinical description of patients, both in the main text (page 6, lines 111-126) and Table 1. We have not found any overlap with encephalocraniocutaneous lipomatosis (ECCL). Ocular manifestations in *GNA13* patients mainly consist of coloboma of the iris or eyelid, not epibulbar dermoids as in ECCL, and neither sebaceous naevus, lipomatosis/naevus psiloliparus nor aplasia cutis were present. However, the *GNA13* phenotype overlaps with the *RHOA* mosaic neuroectodermal syndrome, which is in keeping with a common involvement of the *RHOA/ROCK* pathway. Two patients were indeed referred by clinicians with a suspicion of *RHOA* mosaicism. We have also highlighted this point in the manuscript (page 6, lines 110-111).

7) The authors have a line (page 12, top of page) that “the mosaicism we describe here may protect patients from cancer progression”. This line is misleading to me or I am not understanding. I think they are trying to explain the absence of cancer in their patients despite the fact that R200K is a cancer associated mutation. But there is a great deal of precedent in the mosaic conditions for the presence of oncogenic mutations and the absence of cancer. For example, *PIK3CA* related overgrowth patients rarely develop cancer for reasons that remain unclear. The authors have not tested enough “cellular compartments” in their patients to know where the mutation is present or absent, other than the blood. Also three of their patients remain quite young (one has no age listed), so they cannot yet be certain these individuals may not go on to develop cancer. Can the authors please clarify or change this line?

8) I do appreciate the attempt to discuss cancer risk in HI patients, and the authors should probably refer to previous publications where this has been described. The finding of an oncogenic mutation in HI patients certainly might be an explanation for some of the previous reports of cancer and HI association, and I think the authors should note this, but also note that the cancer risk in HI relatively lower than what one might expect, for reasons that remain mostly unknown.

Response to comments 7 & 8: Indeed, no increased risk of cancer has been reported in hypomelanosis of Ito (page 14, lines 364-365). We initially sought to address the link between somatic activating mutations in cancer and mosaic activating mutations in developmental disorders, but we agree that our sentence was confusing. We have deleted it and replaced it with a sentence highlighting that in most mosaic development syndromes involving oncogenes such as *PIK3CA*, the incidence of cancer is not increased (page 14, lines 365-368).

9) What is the developmental expression of *GNA13* in humans and would it be possible that some “isolated birth defects” such as jejunal atresia could be caused by mosaic *GNA13* mutations, even in the absence of hypomelanosis?

Gastrointestinal (GI) tract atresia appears to be a non-random manifestation in our patients (page 12, lines 297-303). In patients with non-syndromic GI tract atresia, it could indeed be caused by mosaic *GNA13* mutations restricted to the gut tissue. In our opinion, many sporadic birth defects are likely to be due to such localised mosaicism. This hypothesis could be tested on tissues obtained from surgical

procedures. Consistently, a search of the Human Protein Atlas indicates that $G\alpha_{13}$ is highly expressed in the gastrointestinal tract (see: <https://www.proteinatlas.org/ENSG00000120063-GNA13/tissue>).

10) My last comment (and only comment on the cell based functional side of the paper) is that many of the assays (f-actin amount, cell roundness, vinculin expression) have a large amount of variance and a great deal of overlap with the “WT” conditions, despite the significant p-values. Can the authors comment on this? I am guessing it is likely due to variance in GNA13 transfection rates- if so is it possible to correct for this? I am not an expert on in vitro cell based experiments (hence all my comments on clinical aspects), so if this reflects normal variance that is fine, but perhaps a sentence in the discussion would be helpful?

It is absolutely true that single-cell analysis poses a unique challenge due to the significant intrinsic heterogeneity of the cells themselves. We fully recognize this complexity and have adopted rigorous analytical approaches to capture and interpret cellular natural variation. We have utilised a large number of cells for each individual experiment to verify and analyse cell behaviour. We are confident that our results accurately reflect the complexity of our biological system of interest. We have now commented on that in the *Discussion* section of the revised manuscript (pages 12-13, lines 320-321).

Reviewer #3 (Remarks to the Author):

This is an interesting paper reporting on GNA13 activating mutations as an explanation for a subset of patients with the clinical pattern of mosaicism termed Hypomelanosis of Ito.

The authors demonstrate this in 4 patients using exome sequencing in one and targeted sequencing of GNA13 in 3 patients. The R200K mutation has been studied previously in bladder cancer cells and shown to be a hotspot that perturbs YAP-TAZ signalling via Rho-GEF/GTPase. There have been some cellular contexts where GNA13 mutations lead to inactivation rather than activation, and hence the characterisation in cells with a melanocyte lineage are important (10.1074/jbc.AC120.014698).

The majority of the paper investigates the consequence of this mutation in cellular models, principally melanoma cell lines. Key readouts are changes in cellular morphology and expression of proteins visualised and quantified on ICC cells. Comparisons are made between the R200K mutation and a more extensively studied change at codon 226.

The basis for hypopigmentation posited ultimately is the reduction in melanosome transfer, or alternatively reduced numbers in the skin due to impaired migration to the skin.

Points to consider:

Investigation on the effect of the GNA13 R200K change on cell proliferation and YAP TAZ signalling are lacking and would give insights in this cellular context.

We have now added extensive additional experimental data regarding the effect of GNA13 R200K on cell proliferation (Page 10, lines 243-245; Fig. 6d), YAP/TAZ expression (Page 9, lines 225-231; Fig. 6a) and YAP/TAZ signalling pathway (Pages 9-10, lines 232-242; Fig. 6b,c and Supplementary Figures 6 and 7a-c) (see below for more details).

1) What are the basal levels of YAP and TAZ in the cell lines used (SKMEL28 and B16F0)? This can vary and can impact on cytoskeleton remodelling and cell migration – see <https://onlinelibrary.wiley.com/doi/10.1111/pcmr.13013>

As requested by the Reviewer, we have examined the basal levels of YAP and TAZ in all cell types used here by Western blot analysis with an antibody that recognizes both YAP and TAZ proteins (see below, Fig. 6a, Page 19 lines 511-525 in *Methods*, and Page 9 lines 225-231 in *Results* sections of the revised version of the manuscript). We have found that B16-F0 and SK-MEL-28 cells co-express YAP and TAZ but at opposite

levels: B16-F0 cells express very high levels of YAP but low levels of TAZ, whereas SK-MEL-28 cells exhibit low YAP expression and high TAZ expression. These differences in YAP and TAZ expressions are fully consistent with the previously published article mentioned by the Reviewer (Reference # 12: Lui JW, Moore SPG, Huang L, Ogomori K, Li Y, Lang D. YAP facilitates melanoma migration through regulation of actin-related protein 2/3 complex subunit 5 (ARPC5). *Pigment Cell Melanoma Res.* 2022 Jan;35(1):52-65. doi: 10.1111/pcmr.13013. Epub 2021 Sep 1. PMID: 34468072). We have also studied YAP and TAZ expression in primary Normal Human Epithelial Melanocytes (NHEM) (Fig. 6a, Page 9 lines 230-231). Expression levels of both proteins appear more homogenous in this cell type.

Fig. 6a:

2) It would be informative to know the consequence on YAP TAZ signalling when GNA13 R200K are transfected in the cell lines used, as it has been shown that YAP levels can influence the number of focal adhesions in melanoma cells.

It is indeed interesting to determine if the $G\alpha_{13}$ R200K mutation affects YAP and TAZ signalling in our cellular models, especially since it has been reported that $G\alpha_{13}$ R200K induces YAP/TAZ-dependent transcriptional activity in HEK293T but not in NIH3T3 cell line (Reference # 11: 10.1074/jbc.AC120.014698).

To address this, we have first investigated YAP and TAZ nuclear translocation in all three cell types used in our study using immunocytochemistry. We have determined that $G\alpha_{13}$ R200K, as well as the $G\alpha_{13}$ Q226L gain-of-function mutant, induce YAP nuclear translocation in B16-F0 cells, SK-MEL-28 and also NHEM (Pages 9-10 lines 232-242; Fig. 6b and Supplementary Figures 6 and 7a). By contrast, the two $G\alpha_{13}$ mutants failed to induce TAZ nuclear translocation (Page 10 lines 235-236; Fig. 6c and Supplementary Fig. 7b). We believe that these important results indicate that the roles of YAP and TAZ are not necessarily identical, even though both proteins are paralogs. This is in line with growing evidence coming from other studies that show that YAP and TAZ are not redundant in some settings, including in melanoma

(Reference # 12: Lui JW, Moore SPG, Huang L, Ogomori K, Li Y, Lang D. YAP facilitates melanoma migration through regulation of actin-related protein 2/3 complex subunit 5 (ARPC5). *Pigment Cell Melanoma Res.* 2022 Jan;35(1):52-65. doi: 10.1111/pcmr.13013. Epub 2021 Sep 1. PMID: 34468072).

Fig. 6b,c (B16-F0 cells):

Supplementary Fig. 7a,b (SK-MEL-28 cells):

Supplementary Fig. 6 (primary NHEM cells):

Furthermore, in order to determine whether the YAP signalling pathway was functional upon expression of gain-of-functions $G\alpha_{13}$ mutants, we next tested whether *ARPC5*, a specific target gene of YAP involved in actin polymerization and migration (Reference # 12), was induced in our conditions. We first transduced SK-MEL-28 with relevant constructs and obtained about half the cells expressing WT (49.7 % +/- 3.7 %), R200K (47.3 % +/- 5.8 %) or Q226L (48.2 +/- 3.3 %) $G\alpha_{13}$. At the whole population level, our results show that *ARPC5* transcripts were upregulated by both $G\alpha_{13}$ R200K and Q226L mutants (see below, Supplementary Fig. 7c and Page 10 lines 236-242), indicating that the YAP signalling pathway is hyperactivated upon expression of either of the two gain-of-function $G\alpha_{13}$ mutants.

Supplementary Fig. 7c:

3) Do these GNA13 R200k transduced cells proliferate more?

We assessed the effects of the $G\alpha_{13}$ R200K and Q226L mutations on cell proliferation using CellTrace Violet dilution and flow cytometry analysis. Our results revealed that both mutations have no effect on proliferation (see below, Fig. 6d and Page 10 lines 243-245).

Fig. 6d:

4) Have these changes been explored outside a cancer cell line context such as in NHEMs? This may give a closer representation to what happens in embryonic skin development.

We agree with the Reviewer that investigating the observed changes on primary non-cancerous cells would strengthen our findings. We now provide additional data showing that primary Normal Human Epithelial Melanocytes (NHEM) exhibit the same changes in cell morphology (Perimeter, Circularity and Solidity), F-actin, P-MLC and RHOA-GTP contents as those observed in the B16-F0 and SK-MEL-28 melanoma cell lines upon $G\alpha_{13}$ R200K or Q226L variants expression (see below, Supplementary Fig. 3 a-c, Page 7 lines 161-164, Page 8 lines 180-182 and lines 197-200).

Furthermore, we have shown above that NHEM cells express YAP and TAZ at equivalent levels and that $G\alpha_{13}$ R200K and Q226L also induce YAP nuclear translocation in these cells (Fig. 6a, Supplementary Fig. 6, Pages 9-10 lines 230-235). Altogether, our new results indicate that human primary melanocytes behave like the melanocytic cell lines we have also used in our study, for all the readouts we have analysed.

Supplementary Fig. 3:

Minor points;

The bioinformatic pipeline used was not clear – have other genes been excluded in the targeted sequencing of these patients? What variant caller was used, and was deduplication of PCR reads used? The % of reads (21-36%) was higher than expected when compared to other mosaic skin conditions affecting pigment cells (eg <5%).

We agree that bioinformatic methods were not clear enough. Hence, we have added more technical data in the *Methods* (Page 16 lines 414-434) and *Results* (Page 6 lines 127-135) sections of the manuscript. We used classic tools: FastQC to assess the quality of sequencing reads; Trimmomatic to remove sequencing adapters and low-quality bases; the Burrows-Wheeler Aligner to align reads to the human genome reference sequence GRCh37/hg19; Genome Analysis Toolkit (GATK) to perform realignment around insertions and deletions; Picard and the GATK to mark PCR duplicates and collect quality-control, sequencing depth, and coverage metrics.

As previously reported (Kuentz et al. 2017 - PMID: 28151489; Vabres et al. 2019 - PMID: 31570889; Carmignac et al. 2021 - PMID: 33833411), we used in the first place targeted amplicon sequencing in affected skin in all patients to exclude *MTOR* and *RHOA* variants. Patient 2 had paired-exome sequencing (coverage of 210X in affected skin vs 80X in blood): we identified the *GNA13* variant in affected skin (absent in blood, see Table 1). We then used *GNA13* targeted amplicon sequencing in affected skin in all patients (coverage ranging from 2374X to 8487X).

Furthermore, the variant allele fraction (VAF), which is indeed quite high, only reflects the level of mutant cells in a given sample (here the skin), not in the whole body. Indeed, we and others have reported highly variable VAFs in different tissues in a patient on post-mortem findings, be it in Proteus syndrome (10.1002/ajmg.a.37612) or *MTOR*-related hypomelanosis of Ito (10.1111/cge.14511).

Point by point :

REVIEWERS' COMMENTS

Reviewer #1 (Remarks to the Author):

Thank you for addressing the points raised previously. The following remain:

Page 12, paragraph beginning line 304: The authors justify the use of melanocytes derived from skin biopsies and maintain that “diagnosis of mutation testing in G α 13-related mosaic disorders should be primarily performed on a biopsy from affected skin.” There should be an acknowledgement here that other, less invasive sources of DNA might be used, in accordance with their response to reviewers stating “we thank the Reviewer for suggesting we test the presence of the GNA13 mutation in other tissues, as it turns out that buccal epithelium could be a less invasive source of cells for some of these patients”. In clinical practice, there is usually a wider differential diagnosis than G α 13-related mosaic disorders, and (particularly in children, and since wound-healing might be impaired) an initial less invasive test, such as venesection or buccal smear, is preferable, usually covering a range of mosaic skin disorders, some of which can be detected on deep sequencing in blood. If that is negative it is reasonable to proceed to skin biopsy and perhaps more targeted testing.

We agree with the reviewer, especially as we have shown the presence of the mutation in the buccal smear in two patients out of four. We have now modified the manuscript accordingly (see Discussion section).

Page 7, line 156: “either both mutants” should presumably be “either or both mutants”.

Page 9, line: “exhibit similar features than” should be “exhibit features similar to”.

Page 12, line 298: what is meant by “Defects in cell 6”?

Page 12, line 302: the word “Consistently” means “invariably” not “consistent with that”. I suggest changing the wording to “Consistent with that hypothesis...”

Page 13, line 323: “The mechanisms responsible.....remains” should be “The mechanisms responsible.....remain”

Page 20, line 548: “the number...were counted” should be “the number...was counted”

We thank the reviewer for pointing out these mistakes, which we have corrected now.

Reviewer #2 (Remarks to the Author):

Thanks to the authors for addressing all of my concerns and comments. The manuscript is significantly improved, and the results important and exciting to our field. I have no additional comments.

We thank the reviewer for the suggestions that improved the revised version of the manuscript significantly.

Reviewer #3 (Remarks to the Author):

Thank you for doing the additional experiments on normal melanocyte cell lines. This adds to the work and the YAP nuclear localisation seen is also helpful. Overall, I am satisfied with the experiments carried out and I think a strong case is made for the impact of this pathogenic variant in GNA13 in these patients affecting the function of melanocyte cell lines.

It is pleasing that there is some alignment of the outcomes of the melanoma cell lines and the melanocyte cell line for readouts studied, however this must be cautiously interpreted (response to point 4) and some additional clarification is needed:

For Figure 3, it could be argued that the impact of the R200K mutation on circularity and f-actin content is more pronounced in the NHEM cells. Could this be shown in Figure 3 as a “side by side” comparison rather than in Supplementary Figure 3? This could align well with the narrative for the paper that the effects seen on cytoskeleton remodelling are similar in both normal and cancer cell lines as the authors have done in the revised text line 161 page 7.

Indeed, compared to B16-F0, the R200K mutation tends to have more noticeable impacts on circularity and F-actin levels in NHEM. Nonetheless, the main objective of presenting these experiments is to determine whether the mutation affects the cell shape and cytoskeleton in both cancerous cell lines and normal non-cancerous cells. Presenting Figure 3 as a side-by-side comparison would be interesting. However, we would prefer to keep the figure as it is, due to space constraints imposed by the journal. Indeed, if we wish to integrate the NHEM into the same figure, we must reduce the size and clarity of both images (B16-F0 and NHEM) so they can fit together in one image based on the journal’s criteria. Considering that Figure 3 represents our first *in vitro* results, which the rest of the experiments are based upon, we believe it is ideal to highlight it by maintaining the image’s optimal quality and size.

For Figure 6 a, it is good that the western blot of presumed basal levels of YAP and TAZ are indicated. Please show the expression levels using immunoblotting of YAP and TAZ both in cells with GNA13 WT overexpression as well as R200K and Q226L across the 3 cell lines.

We have added these experiments in the Supplementary Figure 8 (in the newly revised version):

Overall, the expression of Gα₁₃ WT as well as Gα₁₃ R200K and Gα₁₃ Q226L do not affect the expression of both YAP and TAZ in the three cell lines.

For Figure 6b, the claim of a lack of change in proliferation in the abstract following GNA13 R200K overexpression holds for a melanoma cell line (B16-F0) however this may be different in other melanoma cell lines (SKMEL28) and particularly normal melanocytes (NHEM). It would be important to show this across 3 cell lines, given that in a number of cancers, YAP hyperactivation leads to cellular proliferation (<https://doi.org/10.1038/s41467-024-46531-1>). In the event of variation across cell types, the interpretation of NHEM cells should be favoured, as the context in the paper is that of melanocytes in human skin development.

We have performed the proliferation assay on the NHEM and the SK-MEL-28 cell lines as we had previously done on the B16-F0 cells:

We observe that expression of the two mutants causes no change in proliferation in any of the cell types. We have added these results in the Supplementary Figures 6b, 7d.

Minor points:

I suggest that given the different types of cell lines now used in the paper, that this is indicated within the figures for the ease of the reader to follow the narrative.

This is a helpful tip. For the sake of clarity, we have now indicated the names of the cell lines in the figures. We have also tried to group preferentially all data obtained in a given cell type in the same figure to help the readers.

For the QPCR data Supp Fig 7c – the gene should be the title of the graph.

This has been done.